# Bayesian Robust Cooperative Multi-Agent Reinforcement Learning Against Unknown Adversaries

**Kiarash Kazari & György Dán**
KTH Royal Institute of Technology, Stockholm, Sweden
{kkazari,gyuri}@kth.se

## Abstract

We consider the problem of robustness against adversarial attacks in cooperative multi-agent reinforcement learning (c-MARL) at deployment time, where agents can face an adversary with an unknown objective. We address the uncertainty about the adversarial objective by proposing a Bayesian Dec-POMDP game model with a continuum of adversarial types, corresponding to distinct attack objectives. To compute a perfect Bayesian equilibrium (PBE) of the game, we introduce a novel partitioning scheme of adversarial policies based on their performance against a reference c-MARL policy. This allows us to cast the problem as finding a PBE in a finite-type Bayesian game. To compute the adversarial policies, we introduce the concept of an externally constrained reinforcement learning problem and present a provably convergent algorithm for solving it. Building on this, we propose to use a simultaneous gradient update scheme to obtain robust Bayesian c-MARL policies. Experiments on diverse benchmarks show that our approach, called BATPAL, outperforms state-of-the-art baselines under a wide variety of attack strategies, highlighting its robustness and adaptiveness. [1]

## 1 Introduction

Cooperative multi-agent reinforcement learning (c-MARL) has achieved remarkable performance in areas such as autonomous driving, 5G networks, robotics, and smart grids (Canese et al., 2021), as it allows agents to learn distributed policies for complex sequential tasks. Nonetheless, the failure or the compromise of even a single agent, either through direct manipulation of its actions or by corrupting its observations, can degrade the overall team performance (Lin et al., 2020), calling for policies that are robust against faults and adversarial attacks.

Existing approaches for obtaining robust policies rely on dataset augmentation or on adversarial training (Gleave et al., 2019; Pattanaik et al., 2017; Havens et al., 2018; Pinto et al., 2017; Phan et al., 2021; Liu et al., 2024a; Li et al., 2024). Dataset augmentation involves introducing one or more adversarial perturbations during training, allowing agents to learn under adversarial and nominal conditions simultaneously (Gleave et al., 2019; Pattanaik et al., 2017; Havens et al., 2018). The alternative approach is based on jointly training the benign and the adversarial agents, typically formulated as a zero-sum Stackelberg game, and a saddle-point equilibrium in policies is sought after (Pinto et al., 2017; Phan et al., 2021; Liu et al., 2024a). These approaches typically yield a single policy optimized for adversarial conditions and thus they are typically suboptimal when all agents are cooperative. Even if the trained policy can maintain a belief about the presence of an adversary, as in Li et al. (2024), robust learning based on saddle-point equilibria against a worst case adversary found using gradient descent has three fundamental limitations.

First, it relies on the assumption of a worst case adversary, which fails to capture adversaries with an objective other than minimizing the team reward as well as non-cooperative behavior due to failure. These may deviate substantially from worst-case attacks (Liu et al., 2024b; Kokolakis et al., 2020), and thus the defender's max–min policy may be far from optimal considering the actual adversarial strategy, resulting in poor team performance.

---

[1] Code available at https://github.com/kiarashkaz/BATPAL

Second, the optimization problem solved is inherently non-convex, and learning algorithms are prone to converge to local stationary points, which may not be globally optimal (Kalogiannis et al., 2022; Fiez et al., 2020; Reddi et al., 2024). As a result, the saddle-point policies are local Stackelberg equilibria (Loftin et al., 2024), potentially far from the equilibrium sought after.

Third, exposure to perturbed versions of a single adversarial policy during training can cause the agents' representation of adversarial dynamics to overfit. Consequently, when faced with a different type of adversarial behavior at deployment, the agents may fail to adapt their policies to the previously learned max–min strategy (Liu et al., 2024a). In such cases, they may not even achieve the minimum performance guarantee that the max–min strategy is theoretically expected to provide.

To address these limitations, we introduce a novel approach for training robust MARL policies that can adapt to a diverse set of adversarial behaviors. Instead of learning a single max–min policy, our approach partitions the set of adversarial policies into disjoint subsets, defined by the range of team reward they would impose, and then computes a policy that is robust to a representative adversarial policy from each subset. Although our approach cannot completely eliminate the problem of local stationary points described above, it mitigates the problem by restricting the search to smaller, isolated feasible sets. Moreover, the subsets are constructed so that adversarial policies in different subsets exhibit distinct behaviors. The defender's MARL policy is then trained to adapt based on its belief of adversarial behavior. Our main contributions are as follows.

(1) We introduce a Bayesian Dec-POMDP game model with a continuum of adversarial types and propose a novel criterion for discretizing the type space to ensure exposure to a diverse set of adversarial policies during training. Based on the perfect Bayesian equilibrium of the game, we formulate the Bayesian regret as the objective to characterize the robustness of a policy.

(2) To compute an equilibrium, we introduce the concept of an *externally-constrained RL* to find the adversarial policies of different types. We propose both a provably convergent algorithm and a practically efficient variant to solve this problem. Building on these, we design an end-to-end adversarial learning framework, termed BATPAL, to derive Bayesian robust c-MARL policies.

(3) Through extensive simulations, we demonstrate the effectiveness of BATPAL in adapting to unseen adversarial policies across various benchmark MARL environments, and show that it consistently outperforms state-of-the-art robust MARL algorithms.

**Related Work:** In robust learning the agent–adversary interaction is modeled as a game, and the agents seek a max–min policy for execution-time robustness. RARL (Pinto et al., 2017) and RARAL (Pan et al., 2019) focus on adversarial disturbances with alternating optimization, while Tessler et al. (2019) and RAP (Vinitsky et al., 2020) study adversarial manipulation of actions. Although effective against worst-case attacks, such approaches can be overly conservative; recent work (Liu et al., 2024b) addresses this by considering non-worst-case adversaries, but in a lifelong learning context.

For execution-time robustness in MARL, M3DDPG (Li et al., 2019) adopts a max–min value function, while RAT (Phan et al., 2020) and RADAR (Phan et al., 2021) consider environments with a subset of adversarial agents. Yuan et al. (2023) and Lee et al. (2025) model budget-limited attacks, and Liu et al. (2024a) studies adjustable, non-worst-case adversaries in two-agent scenarios. Most recently, Li et al. (2024) propose to maintain belief states about what teammates are compromised, but considers a worst case adversary only, leaving agents undefended against unseen adversaries.

## 2 MODEL AND PROBLEM FORMULATION

### 2.1 C-MARL MODEL

We consider a Dec-POMDP $\mathcal{M} = (\mathcal{N}, \mathcal{S}, \{\mathcal{A}^i\}_{i \in \mathcal{N}}, R, P, \{\Omega^i\}_{i \in \mathcal{N}}, \mathcal{O}, \mu, \gamma)$, where $\mathcal{N} = \{1, 2, ..., N\}$ is the set of agents, $\mathcal{S}$ is the set of states, and $\mathcal{A}^i$ and $\Omega^i$ are the set of actions and the set of observations of agent $i$, respectively. We assume that $\mathcal{S}$ and $\mathcal{A}^i$ are finite sets. Furthermore, $R(s_t, \mathbf{a}_t)$, $P(s_{t+1}|s_t, \mathbf{a}_t)$, and $\mathcal{O}(\mathbf{o}_t|s_t)$ denote the reward function, the state transition probability, and the conditional observation probabilities, respectively. Finally, $\mu$ and $\gamma < 1$ denote the initial state distribution and the discount factor, respectively. We denote the history of observations, rewards and own actions of agent $i$ up to time $t$ by $\tau_t^i$.

We assume that the reward is bounded such that, without loss of generality, $|R(s, \mathbf{a})| \leq 1$, $\forall (s, \mathbf{a}) \in \mathcal{S} \times \mathcal{A}$. The value function when the agents follow a joint policy $\boldsymbol{\pi} = (\pi^1, ..., \pi^N)$ is defined as: $V^{\boldsymbol{\pi}}(s) = \mathbb{E}[\sum_{t=0}^{\infty} \gamma^t R(s_t, \mathbf{a}_t) \mid s_0 = s]$. We define the expected initial state value as $V^{\boldsymbol{\pi}} = \mathbb{E}_{s_0 \sim \mu}[V^{\boldsymbol{\pi}}(s_0)]$. Throughout the paper, we use the game-theoretic notation $\mathbf{x}^{-i}$ to denote the collection of $x^j$ for all agents $j \neq i$, where $x$ can be actions, observations, or any other quantity.

## 2.2 BAYESIAN DEC-POMDP AS A MODEL OF ADVERSARIAL ROBUSTNESS

During deployment, agents may deviate from their pre-trained policies due to hardware or software error and due to adversarial activity (Lin et al., 2020; Kazari et al., 2023; 2025). The identity and the objective of non-cooperative agents is, however, unknown to the cooperative agents. Yet, most of the literature on robust single agent and multi agent RL focuses on worst case adversaries, i.e., one that minimizes the team reward (Gleave et al., 2019; Tessler et al., 2019; Li et al., 2019; 2024). Only a few recent works considered robustness to non-worst case adversaries in a single agent setting, e.g., assuming the adversary may not fully control the victim (Liu et al., 2024a), via a population of adversaries (Vinitsky et al., 2020), or via repeated encounters in a bandit setting (Liu et al., 2024b).

To capture diversity of adversarial agents' objectives and the resulting uncertainty, we propose the Bayesian Dec-POMDP defined as

$$\mathcal{M}_B = (\mathcal{N}, \mathcal{S}, \{\Theta^i\}_{i \in \mathcal{N}}, \{\mathcal{A}^i\}_{i \in \mathcal{N}}, R, P, \{\Omega^i\}_{i \in \mathcal{N}}, \mathcal{O}, \mu, \gamma),$$

where $\Theta^i$ is the type space of agent $i$, extending the Dec-POMDP formulation. The type $\Theta^i$ captures the uncertainty about the reward function of agent $i$, and is a subset of $\mathbb{R}^{|\mathcal{S}| \times |\mathcal{A}^i|}$, in general. Nonetheless, as every compact subset of a Euclidean space is equinumerous to $[0, 1]$, we can consider $\Theta^i = [0, 1]$.

The type $\theta^i \in \Theta^i$ of agent $i$ is drawn at the beginning of each episode. We denote by $b_0$ the agents' prior about the types and the initial system state, obtained based on $\mu$. The type $\theta^i = 0$ corresponds to agent $i$ aiming to maximize the team reward, each $\theta^i > 0$ corresponds to an agent that aims to maximize some other reward function. For notational convenience, we use $\theta^i = 1$ as the type of an adversarial agent that aims to minimize the team reward. The policy $\pi^i(a_t^i | \tau_t^i, \theta^i)$ of agent $i$ is thus a function of its type $\theta^i$, and the joint action of the agents has distribution $a_t \sim \Pi_{i \in \mathcal{N}} \pi^i(a_t^i | \tau_t^i, \theta^i)$ and it governs the state transitions. Importantly, even if states are fully observable, the policies of the agents need not be stationary due to incomplete information; they depend on their beliefs $b^i(\tau_t^i)$ about the types of the other agents and their policies, maintained based on the observation history.

Our Bayesian Dec-POMDP formulation generalizes existing formulations in the literature (Li et al., 2024; 2019; Yuan et al., 2023). A Dec-POMDP with only cooperative agents corresponds to $\Theta = 0_N$, mixed cooperative-competitive settings with $N_A$ victim agents correspond to $||\Theta||_0 = N_A$ (Li et al., 2024), and a single worst case adversary corresponds to $||\Theta||_0 = ||\Theta||_1 = 1$.

## 2.3 THREAT MODEL AND PROBLEM STATEMENT

Aligned with the game model $\mathcal{M}_B$ we consider that the identity of the victims and the adversarial objective are unknown to benign agents, and the types of the agents do not change during an episode. For ease of notation we consider that the adversary takes control of a single victim agent $v \in \mathcal{N}$, and we denote the adversarial policy by $\rho^{v, \theta^v} = \pi^v(\cdot | \tau^v, \theta^v)$. Game $\mathcal{M}_B$ is a Bayesian game with imperfect information, its solution is thus a perfect Bayesian equilibrium (PBE), defined as follows.

**Definition 2.1.** *A perfect Bayesian equilibrium (PBE) is a profile of cooperative policies $(\pi^i)_{i \in \mathcal{N}}$ and of adversarial policies $(\rho^{v, \theta_v})_{v \in \mathcal{N}, \theta_v \in \Theta^v}$, and a belief system $(b^i(\tau^i))_{i \in \mathcal{N}}$ and $(b^v(\tau^v, \theta^v))_{v \in \mathcal{N}}$, that satisfies (i) each policy is optimal in expectation at every history given the beliefs (sequential rationality) (ii) beliefs are updated using Bayes rule based on the equilibrium policies for on-path histories, as well as for off-path histories whenever possible.*

To evaluate agent policies, with a slight abuse of notation, let us denote the expected initial state value when non-victim agents follow the joint policy $\boldsymbol{\pi}^{-v}$ and the victim follows policy $\rho^{v, \theta_v}$ by $V^{\boldsymbol{\pi}, \rho^{v, \theta_v}}$. Intuitively, for any victim agent $v$ and policy $\rho^{v, \theta^v}$, the non-victim agents should perform optimally, i.e., as close as possible to the optimal policy against $\rho^{v, \theta^v}$. We can thus evaluate the

policies in terms of the *Bayesian regret* defined as

$$\mathcal{R}(\boldsymbol{\pi}) = \mathbb{E}_{(v,\theta^v)\sim b_0}[\mathcal{R}_{\rho^{v,\theta^v}}(\boldsymbol{\pi})] = \mathbb{E}_{(v,\theta^v)\sim b_0}[\max_{\boldsymbol{\pi}'}(V^{\boldsymbol{\pi}',\rho^{v,\theta^v}}) - V^{\boldsymbol{\pi},\rho^{v,\theta^v}}], \tag{1}$$

where expectation is taken over the prior $b_0$. Observe that a PBE minimizes (1) by definition, and our objective is to learn such an equilibrium policy profile $(\pi^i)_{i\in\mathcal{N}}$.

## 3 BAYESIAN TYPE-PARTITIONED ADVERSARIAL LEARNING (BATPAL)

Ideally, the defender would learn a policy that minimizes the Bayesian regret. However, since the attacker can choose from (possibly infinitely) many adversarial policies, finding a policy that corresponds to a PBE is computationally infeasible. To overcome this issue, we propose a novel approach that partitions the adversarial type space into a finite number of subsets, resulting in a Bayesian Dec-POMDP $\hat{\mathcal{M}}_B$ with type space $\hat{\Theta}^i = \{0, 1, , \ldots, K\}$ for agent $i$. Assuming at most one victim agent, the support of $p$ is the set of all $\hat{\theta}$ such that $||\hat{\theta}||_0 \leq 1$. This set can be equivalently represented by

$$\mathcal{Z} = \{(v, k) : v \in \mathcal{N}, k \in \{1, 2, ..., K\}\} \cup \{\mathbf{0}\}, \tag{2}$$

where $\mathbf{0}$ represents the non-adversarial type for all agents. We denote by $p(\hat{\theta})$ the common prior over $\hat{\theta} = (\hat{\theta}^1, ..., \hat{\theta}^N)$. With a slight abuse of notation, we use both $b^i(\hat{\theta}|\tau^i)$ and $b^i(z|\tau^i)$ for $z \in \mathcal{Z}$ to denote the beliefs.

While partitioning itself is conceptually simple, it is not straightforward how to map adversarial types $\Theta^i$ to $\hat{\Theta}^i$, and how to choose adversarial policies that would be representative for each discrete adversarial type $\hat{\theta}^i$. Our proposed solution is to partition adversarial types based on their severity, defined appropriately, and to use the worst-case policy in each partition as representative. The core idea is then to train a single policy that performs optimally against the worst-case adversarial policies in all partitions, i.e., a PBE of game $\hat{\mathcal{M}}_B$. This approach allows us to explore a rich set of adversarial policies during training, which is essential for obtaining a PBE policy profile.

### 3.1 REFERENCE-VALUE BASED PARTITIONING

The main issue in partitioning the type space is that the reward function for each type $\Theta^v$ is private to agent $v$, and other agent cannot know it. If the other agents were to distinguish between two different types in $\Theta^v$, they only could do it by playing against these two types with a fixed policy and observing the rewards they get. This motivates us to define our partitioning based on how well different adversarial types perform against a reference baseline policy.

Let $\boldsymbol{\pi}_0 \in \arg\max_{\boldsymbol{\pi}} V^{\boldsymbol{\pi}}$ be a cooperative policy profile. We refer to $\boldsymbol{\pi}_0$ as the *reference policy* and denote $V_{\max} = V^{\boldsymbol{\pi}_0}$. Given a victim agent $v$ of type $\theta^v$, let us denote the minimum expected initial state value by $V_{\min}^v = \min_{\rho^v} V^{\boldsymbol{\pi}_0,\rho^v}$. Note that $V_{\min}^v$ is the lowest value an adversarial policy played by $v$ can impose while the other players use $\boldsymbol{\pi}_0$. Thus, under the reference policy of the non-victim agents, the expected initial state value induced by an attack on $v$ lies in $[V_{\min}^v, V_{\max}]$. We can thus define the severity of an adversarial policy $\rho^v$ as

$$\eta_{\rho^v} = \frac{V_{\max} - V^{\boldsymbol{\pi}_0,\rho^v}}{V_{\max} - V_{\min}^v}. \tag{3}$$

The severity of every adversarial policy satisfies $\eta_{\rho^v} \in [0, 1]$ by definition. Essentially, assuming that the adversary plays optimally with respect to its private reward function, $\eta$ provides a mapping from $\Theta^v$ to $[0, 1]$, by which the type 0 (non-adversarial) remains unchanged.

We use the above to partition adversarial policies according to their severity and the victim agent. A policy $\rho^v$ belongs to adversarial type $z = (v, k)$ if $\eta_{\rho^v} \in (\frac{k-1}{K}, \frac{k}{K}]$, and we denote the set of all such policies by $\Pi_z$. Note that any adversarial policy belongs to exactly one of the sets $\Pi_z$ for $z \in \mathcal{Z}$. The following proposition shows that such a partitioning is possible.

**Proposition 3.1.** *If states are observable, then $\Pi_z$ is a nonempty set for all $z \in \mathcal{Z}$.*

Given this partitioning, the discrete adversarial types $\hat{\theta}^v$ correspond to sets $\Pi_z$, and it can be shown that the PBE in $\hat{M}_B$ corresponds a policy $\boldsymbol{\pi}^* = (\pi^{*1}, \ldots, \pi^{*N})$ such that

$$\pi^{*i}(.|\tau^i, \theta^i = 0) \in \arg\max_{\pi^i} \mathbb{E}_{b^i(z|\tau^i)}\Big[\min_{\rho^v \in \Pi_z} V^{\boldsymbol{\pi}^*,\rho^v}\Big], \forall i \in \mathcal{N}, \forall \tau^i. \tag{4}$$

That is, each adversarial policy is a worst case policy in the corresponding partition, and $\boldsymbol{\pi}^*$ minimizes the Bayesian regret (see Appendix B.1 for proof).

Before presenting our solution to (4), we first elaborate on how such categorization enhances the robustness of MARL compared to learning a single max-min policy. First, as empirically showed by Kazari et al. (2023), there is a general trade-off between the impact of an attack and the abnormality of the victim agent's behavior as perceived by non-victim agents. Here, the abnormality refers to a difference between what the non-victim agents expect to observe based on the reference policy and what the victim actually does. Thus, one would expect that if two adversarial policies have a large difference in $V^{\boldsymbol{\pi}_0, \rho^v}$, and accordingly belong to very distinct severity levels, their behavior would be easy to distinguish from the non-victim agents' perspective. This would help MARL training to encounter a diverse set of adversarial policies. The next two propositions provide theoretical support for this reasoning. Proposition 3.2 establishes a bound on the KL divergence between two arbitrary policies in terms of their reference initial state values. The KL divergence quantifies the discrepancy between two probability distributions and is commonly employed as a metric for evaluating policy diversity in regularized reinforcement learning tasks (Yuan et al., 2023; Derek & Isola, 2021).

**Proposition 3.2.** *Consider a victim agent $v$ and any two adversarial policies $\rho^{v,\theta_1}$ and $\rho^{v,\theta_2}$. If states are observable then we have*

$$\mathbb{E}_{s \sim d_\mu^{\boldsymbol{\pi}_0, \rho^{v,\theta_1}}} \left[ D_{\mathrm{KL}}(\rho^{v,\theta_1}(s) || \rho^{v,\theta_2}(s)) \right] \geq \frac{(1-\gamma)^2}{2} |V^{\boldsymbol{\pi}_0, \rho^{v,\theta_1}} - V^{\boldsymbol{\pi}_0, \rho^{v,\theta_2}}|^2, \tag{5}$$

*where $d_\mu^{\boldsymbol{\pi}_0, \rho^v}$ is the discounted state visitation distribution under $(\rho^v, \boldsymbol{\pi}_0^{-v})$.*

Following Cohen et al. (2019), we define the diversity of a set of policies $\{\rho^{v,\theta_k}\}_{k=1}^K$ as $\mathrm{Div}(\{\rho^{v,\theta_k}\}_{k=1}^K) = \frac{1}{K(K-1)} \sum_{i \neq j} \mathbb{E}_s[D_{\mathrm{KL}}(\rho^{v,\theta_i}(s) || \rho^{v,\theta_j}(s))]$. Proposition 3.3 provides a lower bound on the diversity of adversarial policies generated by our partitioning through solving (4).

**Proposition 3.3.** *If $\rho^{*v,k}$ denotes the best response of adversarial type $k$ against $\boldsymbol{\pi}^*$ in (4) then*

$$\mathrm{Div}(\{\rho^{*v,k}\}_{k=1}^K) \geq \frac{(1-\gamma)^2(V_{max} - V_{min}^v)^2(K-2)}{12K} \tag{6}$$

Recall that one of the issues with learning a single max-min policy over the entire set of adversarial policies is its sub-optimality when the c-MARL team faces an arbitrary non-worst case attack. The next proposition demonstrates how the proposed partitioning mitigates this issue.

**Proposition 3.4.** *Let $\hat{\rho}^v \in \Pi_z$ be an arbitrary adversarial policy for some $z = (v, k)$ and $\boldsymbol{\pi}_z^* \in \arg\max_{\boldsymbol{\pi}} \min_{\rho^v \in \Pi_z} V^{\boldsymbol{\pi}, \rho^v}$. Then, assuming fully observable states, we have*

$$\mathcal{R}_{\hat{\rho}^v}(\boldsymbol{\pi}_z^*) \leq \frac{k(V_{max} - V_{min}^v)}{K}. \tag{7}$$

The proof is provided in Appendix C.4. To interpret this result, let us compare the case of $K = 1$ with $K > 1$. Note the case with $K = 1$ is equivalent to learning a max-min policy over the set of all adversarial policies with $v$ as the victim. When $K = 1$, the bound on the regret for any arbitrary adversarial policy can get as large as $V_{\max} - V_{\min}^v$. In contrast, when $K > 1$, (7) gives a severity-dependent bound. In particular, for attacks belonging to lower severity levels, i.e., small $k$, the optimality gap becomes smaller, as the ratio $\frac{k}{K}$ becomes smaller.

Finally, although the issue of getting stuck in local optima remains, our partitioning-based approach improves the likelihood of finding better solutions by restricting the search to a collection of smaller, non-overlapping feasible subsets that together cover the entire feasible space of adversarial policies.

## 4 Robust Learning

### 4.1 Learning Adversarial Policies via Externally Constrained RL

To solve (4), we first focus on solving the inner minimization problem, i.e., for a given $z = (v, k)$, a non-victim policy $\boldsymbol{\pi}$ and reference policy $\boldsymbol{\pi}_0$, find a policy $\rho^{v*} = \arg\min_{\rho^v \in \Pi_z} V^{\boldsymbol{\pi}, \rho^v}$. Since throughout the subsection we focus on a single adversarial policy, we drop superscript $v$ for notational simplicity. Observe that from the adversary's perspective, fixing the non-victim policies to any

given policy induces a specific POMDP (see (21) and (22) in the Appendix for the corresponding reward and transition kernel). Thus, the inner minimization is a constrained problem of the form

$$\min_{\rho} \mathbb{E}_{s\sim\mu}[V_{(1)}^{\rho}(s)]$$
$$\text{s.t.} \quad l \le \mathbb{E}_{s\sim\mu}[V_{(0)}^{\rho}(s)] \le h, \tag{8}$$

where $V_{(1)}^{\rho}$ and $V_{(0)}^{\rho}$ denote the initial state value function of policy $\rho$ when it is applied to two different POMDPs, namely POMDP$_1$ and POMDP$_0$, respectively, and $l$ and $h$ are some real numbers. Here, POMDP$_0$ and POMDP$_1$ refer to POMDPs induced by fixing $\boldsymbol{\pi}_0$ and $\boldsymbol{\pi}$, respectively, and $l$ and $h$ are the boundaries of severity type $k$. For notational simplicity, in the rest of the subsection we consider that states are observable, hence we refer to these as MDP$_1$ and MDP$_0$.

Observe that MDP$_1$ and MDP$_0$ share the same action and state spaces, but differ in the reward function and the transition dynamics. Thus, although problem (8) resembles the constrained RL problem in the context of safe learning (Paternain et al., 2019; Liu et al., 2020), there is a fundamental difference. In constrained RL, the costs that define the constraints are essentially obtained through the same trajectory as the rewards in the objective function. On the contrary, in our problem, the objective and the constraints correspond to different MDPs, and consequently, different trajectories. To highlight this difference, henceforth we refer to (8) as the *externally* constrained RL problem.

We propose to use the log barrier method to approximate (8) via an unconstrained problem. That is, we define $V_{(j)}^{\rho} = \mathbb{E}_{s\sim\mu}[V_{(j)}^{\rho}(s)]$ for $j = 0, 1$, and obtain

$$\min_{\rho} \quad V_{(1)}^{\rho} - \lambda \log(V_{(0)}^{\rho} - l) - \lambda \log(h - V_{(0)}^{\rho}), \tag{9}$$

where $\lambda$ is a hyperparameter controlling the optimality-feasibility trade off. We propose to solve (9) using a gradient descent approach on policy $\rho_{\psi}$ parametrized by parameter vector $\psi$. The gradient of the objective function can then be expressed as follows.

**Proposition 4.1.** *The policy gradient of the objective function (9) is*

$$g_{\psi} = \frac{1}{1-\gamma}\mathbb{E}_{s\sim d_{(1)},\, a\sim\rho_{\psi}(.|s)}[\nabla_{\psi}\log\rho_{\psi}(a|s)A_{(1)}^{\rho_{\psi}}(s,a)]$$
$$-\frac{\lambda}{1-\gamma}\Big(\frac{1}{\mathbb{E}_{s\sim\mu}[V_{(0)}^{\rho_{\psi}}(s)]-l} - \frac{1}{h-\mathbb{E}_{s\sim\mu}[V_{(0)}^{\rho_{\psi}}(s)]}\Big)\mathbb{E}_{s\sim d_{(0)},\, a\sim\rho_{\psi}(.|s)}[\nabla_{\psi}\log\rho_{\psi}(a|s)A_{(0)}^{\rho_{\psi}}(s,a)]$$
$$\tag{10}$$

*where $A_{(j)}^{\rho_{\psi}}$ and $d_{(j)}$ denote the advantage function and the discounted state visitation distribution under $\rho_{\psi}$ corresponding to MDP$_j$, respectively. (Proof in Appendix C.5)*

Then, the stochastic update rule for the policy parameters would be

$$\psi_{n+1} = \psi_n - \alpha_n \hat{g}_{\psi_n}, \tag{11}$$

where $\hat{g}_{\psi_n}$ is an estimate of $g_{\psi_n}$ and $\alpha_n$ is the learning rate. To estimate the gradient $g_{\psi}$, let $\hat{V}_{(j)}^{\psi_n}$ and $\hat{\nabla}_{(j)}^{\psi_n}$ denote some unbiased estimators of $V_{(j)}^{\rho_{\psi_n}}$ and $\nabla_{\psi}V_{(j)}^{\rho_{\psi_n}}$, respectively, where $j = 0, 1$. Then, our proposed estimate is

$$\hat{g}_{\psi_n} = \frac{1}{1-\gamma}\left[\hat{\nabla}_{(1)}^{\psi_n} - \lambda\Big(\frac{1}{\hat{V}_{(0)}^{\psi_n} - l} - \frac{1}{h - \hat{V}_{(0)}^{\psi_n}}\Big)\hat{\nabla}_{(0)}^{\psi_n}\right]. \tag{12}$$

To obtain the estimates, we collect trajectories of $M$ episodes in the form $\{(s_{t,m,(j)}, a_{t,m,(j)}, r_{t,m,(j)})_{t=0}^{T_m-1}\}_{m=1}^{M}$ by executing $\rho_{\psi}$ on MDP$_j$ for $j \in \{0, 1\}$. In practice, $T_m$ could be the time to reach a terminal state or the episodic time limit. We propose to maintain two parametrized functions, namely $V_{\phi_{(0)}}$ and $V_{\phi_{(1)}}$, as the critics to estimate $V_{(0)}^{\rho_{\psi}}(s)$ and $V_{(1)}^{\rho_{\psi}}(s)$. Then, $\hat{\nabla}_{(j)}^{\psi}$ is obtained in the same way as a standard actor-critic algorithm (Sutton & Barto, 2018), using the empirical average of $\nabla_{\psi}\log\rho_{\psi}(a|s)A_{\phi_{(j)}}(s,a)$, where $A_{\phi_{(j)}}$ is the advantage function calculated based on $V^{\phi_{(j)}}$. Moreover, we can obtain $\hat{V}_{(0)}^{\psi_n}$ as

$$\frac{1}{M}\sum_{m=1}^{M}\left[\Big(\sum_{t=0}^{T_m-1}\gamma^t r_{t,m,(0)}\Big) + V_{\phi_{(0)}}(s_{T_m,m,(0)})\right]. \tag{13}$$

A major difference between the proposed stochastic update and standard policy gradient methods is that our estimate of the gradient is not unbiased, as $\mathbb{E}[\hat{g}_{\psi_n}] \neq g_{\psi_n}$, even if $\hat{V}_{(0)}^{\rho_{\psi_n}}$ and $\hat{\nabla}_{(j)}^{\psi_n}$ are unbiased estimators. Thus, our algorithm is not guaranteed to converge using standard arguments (Robbins & Monro, 1951). Yet, it does converge with a proper selection of the step sizes, as we show next.

**Proposition 4.2.** *Assume that for a parameterization $\psi$, the following conditions hold:*

*(1) $\hat{V}_{(0)}^{\psi}$ and $\hat{\nabla}_{(j)}^{\psi}$ are unbiased estimators.(2) For any $s \in \mathcal{S}$ and $a \in \mathcal{A}$, the function $\log \rho_\psi(a|s)$ is twice differentiable with respect to $\psi$, and both its first and second derivatives are bounded. (3) There exists a strictly feasible starting point $\psi_0$, i.e., $l < V_{(0)}^{\rho_{\psi_0}} < h$. (4) There exists a constant $\zeta > 0$, such that $\nabla_\psi V_{(0)}^{\rho_{\psi_n}}$ is nonzero when $h - \zeta \leq V_{(0)}^{\rho_{\psi_n}} \leq h$ or $l \leq V_{(0)}^{\rho_{\psi_n}} \leq l + \zeta$.*

*Then, for any $\epsilon, \delta > 0$, there exists a sequence of adaptive step sizes $\{\alpha_n\}$ and some values $N_{iter}$ and $M$, such that after $N_{iter}$ iterations of (11) using (12), we have $\min_{n \leq N_{iter}} \|g_{\psi_n}\| \leq \epsilon$ with probability at least 1-$\delta$. Moreover, as $\lambda \to 0$ the obtained point approaches a KKT point of the constrained problem (8) with probability at least 1-$\delta$. (Proof and detailed expressions of $\alpha_n$, $N_{iter}$ and $M$ are available in Appendix C.6)*

Despite the above convergence result, using (12) poses two practical challenges. First, computing the adaptive step size $\alpha_n$ is computationally expensive and requires estimating bounds on the gradient of $\log \rho_\psi$, which is difficult in general. Second, estimating log-barrier gradients near the boundary of the feasible region is sensitive to noise (Usmanova et al., 2024). Mitigating the sensitivity requires a large number of episodic samples $M$, which is infeasible in practical RL settings.

To address these issues, we propose to incorporate the PPO loss function (Schulman et al., 2017) into the policy updates. The key intuition is that the clipping mechanism in PPO constrains policy updates by preventing large deviations from the current policy. This implicitly mitigates the risk of crossing into the infeasible region due to high-variance gradient estimates, while also eliminating the need to compute adaptive step sizes in practice. Moreover, when the initial policy lies outside the feasible region, the update direction is reversed such that the gradient step encourages convergence toward the feasible set. Then the gradient calculation of our proposed algorithm, which we refer to as *externally-constrained PPO (EC-PPO)*, can be summarized as

$$\hat{g}_{\psi_n}^{\text{EC-PPO}} = \begin{cases} \nabla_{(1)}^{\text{PPO},\psi_n} - \lambda(\frac{1}{\hat{V}_{(0)}^{\psi_n}-l} - \frac{1}{h-\hat{V}_{(0)}^{\psi_n}})\hat{\nabla}_{(0)}^{\psi_n}, & \text{if } l+\zeta \leq \hat{V}_{(0)}^{\psi_n} \leq h-\zeta \\ sign(\hat{V}_{(0)}^{\psi_n} - \frac{1}{2}(l+h))\hat{\nabla}_{(0)}^{\psi_n}, & \text{otherwise} , \end{cases} \tag{14}$$

where $\nabla_{(1)}^{\text{PPO},\psi_n}$ is the gradient of the PPO objective function (Eq.(7) in Schulman et al. (2017)), and $\zeta > 0$ is a small value to prevent gradient explosion.

## 4.2 BAYESIAN ADVERSARIAL MARL TRAINING

To find the perfect Bayesian equilibrium policies in (4), recall that PBE policies are optimal in expectation given the beliefs. Thus, we incorporate $b^i$ as an input to $\pi^i$, represented as $\pi^i(\cdot|\tau^i, b^i, \theta^i = 0)$.

It can be shown that $\hat{\mathcal{M}}_B$ is equivalent to a partially observable stochastic game $\mathcal{G}$ with $N+1$ players, where player $N+1$ plays adversarially against the others (details in Appendix B). This interpretation allows us to employ the framework of adversarial training with min-oracle (Kalogiannis et al., 2022; Liu et al., 2024a) for policy updates. Let $\pi_{\omega^i}(\cdot|\tau_i, b_i)$ be a policy parametrized by $\omega^i$ that represents $\pi^i(\cdot|\tau^i, b^i, \theta^i = 0)$. Also, let $\rho_\psi$ be the adversarial policy parametrized by $\psi = (\psi^z)_{z \in \mathcal{Z} \backslash \mathbf{0}}$, such that $\psi^z$ corresponds to $\pi^v(\cdot|s, \theta^v = k)$ for $z = (v, k)$. Note that if $\bar{V}^{\omega,\psi}$ represents the expected initial state value function of non-adversarial agents in $\mathcal{G}$, their objective is to find $\arg\max_\omega \min_\psi \bar{V}^{\omega,\psi}$. Then, assuming that there is an oracle that for any given policy $\pi_\omega$ returns a best response policy $\psi^*(\omega) = \arg\min_\psi \bar{V}^{\omega,\psi}$, the MARL policy is updated as

$$\omega_{n+1} = \omega_n + \beta_n \nabla_\omega \bar{V}^{\omega_n, \psi^*(\omega_n)}, \tag{15}$$

where $\beta_n$ is a step size. It is straightforward to verify that minimizing $\bar{V}$ is equivalent to minimizing $V$, the expected initial state value function of the original game $\hat{\mathcal{M}}_B$. Consequently, our externally constrained RL algorithm can serve as an oracle to compute $\psi^{z*}$ for each $z \in \mathcal{Z}$, since fixing $\omega$

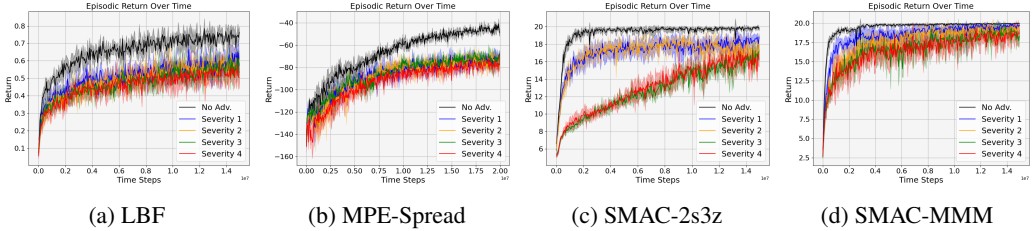

(a) LBF      (b) MPE-Spread      (c) SMAC-2s3z      (d) SMAC-MMM

Figure 1: Average episodic return of the proposed adversarial training, evaluated over 5 runs.

reduces the problem to 8. Moreover, note that for a fixed $\psi$ and assuming updated beliefs, the problem reduces to the standard c-MARL setting.

Such policy optimization is theoretically guaranteed to converge to a Nash equilibrium of Markov games under simplified settings, such as direct parameterization and fully observable states (Kalogiannis et al., 2022; Daskalakis et al., 2020). However, these guarantees rely on performing exact minimization at each policy update and having access to exact gradients, both of which are infeasible in practice. To address this, we employ simultaneous gradient updates, also known as two-timescale stochastic simultaneous gradient descent–ascent (Daskalakis et al., 2020). Then assuming that $\hat{g}_{\boldsymbol{\omega}}(\boldsymbol{\omega}, \boldsymbol{\psi})$ is an unbiased stochastic estimate of $\nabla_{\boldsymbol{\omega}} \bar{V}^{\boldsymbol{\omega}_n, \boldsymbol{\psi}^*(\boldsymbol{\omega}_n)}$, and that $\hat{g}_{\boldsymbol{\psi}}^{\text{EC-PPO}}(\boldsymbol{\omega}, \boldsymbol{\psi})$ is the adversarial gradient derived by (14), our policy updates can be summarized as

$$\boldsymbol{\psi}_{n+1} = \boldsymbol{\psi}_n - \alpha_n \hat{g}_{\boldsymbol{\psi}}^{\text{EC-PPO}}(\boldsymbol{\omega}_n, \boldsymbol{\psi}_n) \tag{16}$$

$$\boldsymbol{\omega}_{n+1} = \boldsymbol{\omega}_n + \beta_n \hat{g}_{\boldsymbol{\omega}}(\boldsymbol{\omega}_n, \boldsymbol{\psi}_n). \tag{17}$$

The intuition is that by selecting $\alpha_n \geq \beta_n$, the adversary's policy serves as an approximate min-oracle, while from the adversary's perspective the c-MARL policy appears nearly quasi-static. To compute $\hat{g}_{\boldsymbol{\omega}}(\boldsymbol{\omega}_n, \boldsymbol{\psi}_n)$, we first need to obtain the agents' beliefs. For this purpose, we employ a parametrized function approximator $b_{\chi_i}(\theta^{-i}|\tau^i)$, implemented using a Recurrent Neural Network (RNN) that takes $\tau^i$ as input. The belief model is trained against the true type $\theta^{-i}$ using a cross-entropy loss. Then, by feeding $(b^i, \tau^i)$ into the policy network and using the value estimate provided by the critic $V_{\phi(1)}(\bar{s})$, we compute $\hat{g}_{\boldsymbol{\omega}}(\boldsymbol{\omega}_n, \boldsymbol{\psi}_n)$ in the same way as in a standard actor-critic algorithm. We refer to our algorithm as Bayesian Type-Partitioned Adversarial Learning (BATPAL), and provide its pseudo-code in the Appendix.

## 5 EVALUATION

We evaluate BATPAL against various attack types in four c-MARL environments. We consider the 2s3z and MMM scenarios from the StarCraft II Multi-Agent Challenge (SMAC) (Samvelyan et al., 2019), with five and ten agents, respectively. We use scenario (10x10-5p-10f-c) in Level-Based Foraging (LBF) (Papoudakis et al., 2021) and the Spread scenario from Multi-Particle Environments (MPE) (Mordatch & Abbeel, 2017), involving five and three agents, respectively. Additionally, we include results for SMAC scenarios 1c3s5z and 11m in the Appendix.

In all environments, we applied our algorithm to train a robust c-MARL policy and a set of adversarial policies with different severity indices. We used MAPPO (Yu et al., 2022) both for updating the c-MARL policy in adversarial learning and to obtain the reference policy $\boldsymbol{\pi}_0$ in the pre-training phase. We assumed a uniform prior over all possible types in the training. Moreover, for a low-complexity implementation, we used parameter-sharing across all agents. Accordingly, we maintained a single neural network for c-MARL policy and $K$ networks for different adversarial types. Moreover, during training, we update adversarial networks in a randomized way, i.e., only one of $K$ networks is updated per each update of the c-MARL policy (see the pseudo code in Appendix E.1) For more details on implementation, we refer to Appendix E.2.

**Baselines** We compare our proposed method with state-of-the-art baselines including EIR-MAPPO (Li et al., 2024), Generalized Maxmin (Gen-Maxmin) (Liu et al., 2024a), and RAP (Vinitsky et al., 2020). We also include the evaluation of the vanilla MAPPO algorithm against the considered attacks. Moreover, to provide a comprehensive assessment of the results, we include for each

**LBF**

| | Ours | EIR-M | GenM | RAP | MAPPO | KT |
|---|---|---|---|---|---|---|
| No Attack | **1.00** | 0.59 | 0.79 | 0.71 | 0.97 | 1.00 |
| Severity 1 | **0.68** | 0.41 | 0.47 | 0.26 | 0.38 | 0.85 |
| Severity 2 | **0.50** | 0.18 | 0.29 | 0.24 | 0.21 | 0.68 |
| Severity 3 | **0.32** | 0.12 | 0.29 | 0.26 | 0.12 | 0.59 |
| Severity 4 | **0.35** | 0.00 | 0.18 | 0.21 | 0.06 | 0.62 |
| A-EIR-MAPPO | **0.26** | 0.06 | 0.03 | 0.15 | 0.00 | 0.35 |
| A-Gen-Maxmin | 0.38 | 0.21 | **0.68** | 0.29 | 0.18 | 0.68 |
| A-RAP | **0.32** | 0.15 | 0.26 | 0.15 | 0.24 | 0.53 |
| ACT | **0.35** | 0.15 | 0.24 | 0.26 | 0.24 | 0.50 |
| DYN-1 | **0.53** | 0.32 | 0.44 | 0.18 | 0.24 | 0.71 |
| DYN-2 | **0.59** | 0.24 | 0.50 | 0.24 | 0.35 | 0.53 |

**MPE-Spread**

| | Ours | EIR-M | GenM | RAP | MAPPO | KT |
|---|---|---|---|---|---|---|
| No Attack | **1.00** | 0.97 | 0.94 | 0.57 | 0.99 | 1.00 |
| Severity 1 | **0.87** | 0.78 | 0.83 | 0.58 | 0.85 | 0.84 |
| Severity 2 | **0.81** | 0.57 | 0.70 | 0.17 | 0.70 | 0.83 |
| Severity 3 | **0.77** | 0.62 | 0.68 | 0.31 | 0.67 | 0.82 |
| Severity 4 | **0.73** | 0.49 | 0.66 | 0.02 | 0.58 | 0.82 |
| A-EIR-MAPPO | 0.64 | **0.67** | 0.47 | 0.30 | 0.54 | 0.69 |
| A-Gen-Maxmin | 0.80 | 0.49 | **0.83** | 0.02 | 0.61 | 0.83 |
| A-RAP | 0.61 | **0.65** | 0.54 | 0.24 | 0.55 | 0.66 |
| ACT | **0.81** | 0.49 | 0.66 | 0.00 | 0.53 | 0.83 |
| DYN-1 | **0.67** | 0.51 | 0.51 | 0.35 | 0.59 | 0.77 |
| DYN-2 | **0.72** | 0.47 | 0.62 | 0.05 | 0.63 | 0.73 |

**SMAC-2s3z**

| | Ours | EIR-M | GenM | RAP | MAPPO | KT |
|---|---|---|---|---|---|---|
| No Attack | 0.98 | 0.96 | **0.98** | 0.94 | 0.96 | 1.00 |
| Severity 1 | 0.66 | 0.49 | **0.84** | 0.79 | 0.58 | 0.96 |
| Severity 2 | **0.55** | 0.12 | 0.18 | 0.39 | 0.11 | 0.94 |
| Severity 3 | **0.60** | 0.09 | 0.00 | 0.09 | 0.00 | 0.73 |
| Severity 4 | **0.70** | 0.07 | 0.05 | 0.08 | 0.00 | 0.73 |
| A-EIR-MAPPO | **0.38** | 0.20 | 0.04 | 0.17 | 0.00 | 0.65 |
| A-Gen-Maxmin | 0.50 | 0.15 | **0.64** | 0.47 | 0.08 | 0.69 |
| A-RAP | 0.51 | 0.19 | 0.01 | **0.58** | 0.00 | 0.64 |
| ACT | **0.72** | 0.35 | 0.22 | 0.20 | 0.00 | 0.75 |
| DYN-1 | **0.52** | 0.27 | 0.12 | 0.21 | 0.10 | 0.83 |
| DYN-2 | 0.71 | 0.56 | 0.57 | **0.74** | 0.38 | 0.90 |

**SMAC-MMM**

| | Ours | EIR-M | GenM | RAP | MAPPO | KT |
|---|---|---|---|---|---|---|
| No Attack | **1.00** | 0.98 | **1.00** | **1.00** | **1.00** | 1.00 |
| Severity 1 | 0.96 | 0.86 | **0.97** | 0.95 | 0.79 | 1.00 |
| Severity 2 | 0.88 | 0.74 | **0.89** | 0.80 | 0.59 | 0.96 |
| Severity 3 | **0.85** | 0.83 | 0.62 | 0.82 | 0.40 | 0.92 |
| Severity 4 | 0.78 | 0.61 | **0.79** | 0.74 | 0.55 | 0.84 |
| A-EIR-MAPPO | **0.85** | 0.83 | 0.62 | 0.80 | 0.43 | 0.87 |
| A-Gen-Maxmin | **0.93** | 0.62 | 0.92 | 0.83 | 0.51 | 0.98 |
| A-RAP | **0.93** | 0.82 | 0.49 | 0.84 | 0.37 | 0.93 |
| ACT | **0.89** | 0.75 | 0.79 | 0.78 | 0.46 | 0.87 |
| DYN-1 | **0.82** | 0.62 | 0.70 | 0.78 | 0.47 | 0.90 |
| DYN-2 | **0.92** | 0.80 | 0.86 | 0.83 | 0.42 | 0.96 |

Figure 2: Performance in four environments against 10 adversarial policies (**Best**, 2nd best not considering KT). The episode rewards for MPE-Spread and LBF are in [-189, -47] and [0.4, 0.74].

attack the results obtained using an oracle defender that is aware of the type of the adversary and is trained against it. This baseline, referred to as Known Type (KT), serves as an empirical upper bound. Finally, we include a comparison with ROMANCE (Yuan et al., 2023) in Appendix D.2.

**Attacks** We use 10 adversarial policies for the evaluation. On the one hand, the policies trained in the adversarial training process of BATPAL, these are indexed by their severity level. In addition, we use the adversarial policies trained against EIR-MAPPO, Gen-Maxmin and RAP, these are marked as "A-X," where X corresponds to the name of the baseline. To assess generalization, we further evaluate all methods against three dynamic adversaries, unseen by all methods. These adversaries are trained by fixing non-victim policies and training an RL agent with a reward function that balances adversarial impact on c-MARL performance with detectability (Kazari et al., 2023). We consider three such adversaries: ACT, which minimizes team reward, and DYN-1 and DYN-2, with DYN-2 placing greater emphasis on close to normal behavior (low detectability). For all attacks, we apply the policy to any victim agent for 50 episodes and report the averages across all episodes.

## 5.1 RESULTS

Figure 1 shows the learning curves of BATPAL with $K = 4$ severity levels. The curves show the episodic rewards of the learned policy when evaluated in both non-adversarial settings and against simultaneously trained adversarial policies. The results demonstrate the convergence of the proposed training scheme across all scenarios and adversarial types.

Figure 2 compares the performance of BATPAL with baselines. For SMAC environments, we use the team win rate of the c-MARL agents as the performance metric, while for the other environments we use the mean episodic total reward, normalized to enable unified comparison across environments. We can make several key observations based on the results. First, in terms of non-adversarial performance, the BATPAL performs at least as well as vanilla MAPPO, indicating that robustification does not compromise optimality under normal conditions. Second, although we train a single cooperative policy profile, it almost always outperforms the robust baselines policies when they face the attack they are trained against. This highlights the importance of exposing agents to a diverse set of adversarial policies in order to obtain robust policies. Third, from the adversary's perspective, the worst performance of other baselines in many cases occurs when they face one of the attacks generated for training BATPAL, rather than their own adversarial policies. This can be attributed to adversarial training getting stuck in local stationary points, which further justifies our proposed method for adversarial search over disjoint sets. Fourth, although the upper bound represented by KT is obtained empirically and may not correspond to the true upper bound on the performance of a robust policy, the performance gap to KT provides an indication of the regret associated with each policy. In many cases, our algorithm achieves near no-regret, even against unseen attacks. It is also worth noting that, even for previously encountered attacks, uncertainty regarding both the

Table 1: Results of ablation study on the MPE-Spread and SMAC-2s3z environments.

(a) MPE-Spread

| Variation | No Attack | ACT | DYN 1 | A-Gen-Maxmin |
|---|---|---|---|---|
| BATPAL | 1.00 | 0.81 | 0.67 | 0.80 |
| No Belief | 0.81 | 0.70 | 0.67 | 0.61 |
| Perfect Belief | 1.02 | 0.75 | 0.78 | 0.68 |
| EC PG | 0.97 | 0.75 | 0.65 | 0.74 |
| Fixed Types | 0.95 | 0.78 | 0.59 | 0.67 |

(b) SMAC-2s3z

| Variation | No Attack | ACT | DYN 1 | A-Gen-Maxmin |
|---|---|---|---|---|
| BATPAL | 0.98 | 0.72 | 0.52 | 0.50 |
| No Belief | 0.96 | 0.33 | 0.30 | 0.36 |
| Perfect Belief | 1.00 | 0.34 | 0.42 | 0.39 |
| EC PG | 0.96 | 0.41 | 0.38 | 0.41 |
| Fixed Types | 0.94 | 0.42 | 0.14 | 0.19 |

adversary's type and the identity of the victim agent (if any) prevents the defender from consistently executing an optimal policy. We provide more results in the Appendix.

## 5.2 ABLATION STUDY

In this section we evaluate the importance of various components of BATPAL by replacing each of them with some alternatives. Table 1 shows the results of evaluating different variations of our method against three representative unseen attacks and also in the non-adversarial setting in MPE-Spread and SMAC-2s3z.

**Beliefs** We considered two variants of BATPAL to evaluate the effect of the belief network. The first variant is trained without a belief, while in the second variant the true type of the adversary is provided directly to the policy network instead of a belief estimate. In Table 1, these two variants are labeled *No Belief* and *Perfect Belief*, respectively. The results show that without the belief, the robustness of the c-MARL policy trained using BATPAL is significantly worse than when using the belief. In this case, even the performance in the non-adversarial setting can be affected as the results in MPE-Spread show. Interestingly, when the perfect belief is fed to the policies, robustness still suffers when facing attacks unseen during training. This is due to that the belief captures the severity level of the attack, but not the actual attack type. These results clearly show the importance of using the belief in BATPAL.

**EC RL Algorithm** To evaluate the effect of PPO in the externally constrained updates of adversarial policies, we consider a variant in which the adversarial updates are performed using a simple policy gradient method without the clipping mechanism. We refer to this approach as EC PG. As shown in Table 1, although this method achieves performance close to BATPAL in non-adversarial settings, it is not generally robust against attacks with different adversarial types. This can be explained by the fact that, without the clipping mechanism, the policies can easily move far outside the feasible region. Consequently, the adversarial policies encountered during training are not representative of their designated types, which undermines overall robustness.

**Ensemble of Adversarial Types** Finally, to evaluate the effect of the proposed partitioning method, we consider an alternative approach in which the c-MARL policies are trained against an ensemble of fixed adversarial policies. For this approach, we use four dynamic adversaries (Kazari et al., 2023) trained with different trade-offs between severity and detectability to ensure diversity among the adversaries used during training. Table 1b shows that this approach, called *Fixed Types*, can lead to significantly lower performance compared to BATPAL. This observation highlights that while diversity is necessary, it is not sufficient to achieve generalizable robustness against unknown adversaries.

## 6 CONCLUSION

We showed that reference-value–based partitioning of adversarial types enhances the adaptability of c-MARL agents to unseen adversaries by exposing the c-MARL team to a diverse set of adversarial policies, demonstrated both theoretically and empirically. We proposed EC-PPO to learn adversarial policies of different types and demonstrated that it can be effectively integrated into our Bayesian adversarial learning framework BATPAL. Our results show that BATPAL outperforms the state-of-the-art by achieving almost no-regret performance against various unseen attacks.

ACKNOWLEDGMENTS

This work was supported in part by the Swedish Research Council under grants 2020-03860, 2024-05269 and 2025-06310 and by the Swedish Civil Defence and Resilience Agency (Project MAD-VAMCHS). The computations were enabled by resources provided by the National Academic Infrastructure for Supercomputing in Sweden (NAISS) at Linköping University partially funded by the Swedish Research Council through grant agreement no. 2022-06725.

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

APPENDIX

## A    RELATED WORK

Adversarial robustness in reinforcement learning has been studied mainly through two main approaches: adversarial training and robust learning. In adversarial training, a known form of adversarial perturbation is introduced during the training phase, allowing the agent to learn both adversarial and nominal transitions simultaneously. For example, Gleave et al. (2019); Pattanaik et al. (2017) employ this approach to defend against various types of manipulations. A closely related method is (Havens et al., 2018) which applies adversarial training within a meta-learning framework to enable adaptation to attacks. These methods primarily target training-time attacks and rely on prior knowledge of the adversary.

Robust learning instead models the agent–adversary interaction as a game, often zero-sum, where the agent seeks a max–min policy for execution-time robustness. Such works are often categorized under robustness to uncertainty, but the uncertainty is explicitly modeled as being induced by an adversary. For instance, RARL (Pinto et al., 2017) and RARAL (Pan et al., 2019) consider adversaries capable of applying model disturbances and propose alternating optimization methods to find a robust policy. On the other hand, Tessler et al. (2019) and RAP (Vinitsky et al., 2020) study adversarial manipulation of actions. Although effective against worst-case attacks, such approaches can be overly conservative; recent work (Liu et al., 2024b) addresses this by considering arbitrary non-worst-case adversaries in a lifelong learning context.

In MARL, robustness has been explored both at training and at execution. Training-time defenses include adversarial regularization for smooth policies (Bukharin et al., 2023) and consensus-based learning robust to Byzantine agents (Ye et al., 2024). Execution-time resilience has been studied through robust learning. M3DDPG (Li et al., 2019) adopts a max–min value function with the idea that each agent assumes all other agents to be adversarial. RAT (Phan et al., 2020) and RADAR Phan et al. (2021) consider environments with a subset of worst-case adversarial agents. ROMANCE (Yuan et al., 2023) and WALL (Lee et al., 2025) consider adversaries capable of targeting all agents but with a limited budget of attack numbers. Recently, Li et al. (2024) propose adversarial belief states that allow agents to adapt online when teammates are compromised. While this approach addresses the challenge of reacting to attacks on different agents, it remains focused on worst-case robustness and does not capture the diversity of adversarial strategies. Finally, Liu et al. (2024a) studies adaptation to non-worst-case adversaries in two-agent scenarios. However, the authors consider a fixed adversarial, which limits the generalization of the robustness to unseen attacks.

## B    BAYESIAN GAME FORMULATION

### B.1    DISCRETE TYPE FORMULATION

As explained in Section 3, $\hat{\mathcal{M}}_B$ is a Bayesian game with type spaces $\hat{\Theta}^i$. In Bayesian games the strategy space is assumed to be type-independent, while the utility is assumed to be type dependent. To align our problem with this model we define the utility function as

$$u^i(\pi^i, \boldsymbol{\pi}^{-i}, \hat{\theta}) = \begin{cases} V^{\boldsymbol{\pi}}, & \text{if } \hat{\theta}^i = 0 \\ -V^{\boldsymbol{\pi}}, & \text{if } \hat{\theta}^i = k, \pi^i \in \Pi_z, z = (i, k) \in \mathcal{Z} \\ -\infty, & \text{if } \hat{\theta}^i = k, \pi^i \notin \Pi_z, z = (i, k) \in \mathcal{Z} \end{cases}$$

The last line is to restrict the set of adversarial policies of each type to the corresponding set $\Pi_z$, and the second line is based on our assumption that the representative of each discrete type is the worst case adversarial strategy in the corresponding partition.

Now assume that $(\boldsymbol{\pi}^*, \boldsymbol{\rho}^*)$ with $\rho^{*v} = (\rho^{*v, \hat{\theta}_1^v}, \dots, \rho^{*v, \hat{\theta}_K^v})$ is a PBE of $\hat{\mathcal{M}}_B$. With the utilities defined above, it can be immediately seen that it corresponds to a solution of (4) and vise-versa. This is because when player $v$'s type is $\hat{\theta}_k \neq 0$, it knows its own type and also the type of the other players. So for a fixed strategy profile $\boldsymbol{\pi}^*$, it plays a strategy $\rho^{v, \hat{\theta}_k^v}$ that minimizes $V^{\boldsymbol{\pi}^*, \rho^{v, \hat{\theta}_k^v}}$ within the corresponding set $\Pi_z$. On the other hand, when a player $i$'s type is $\hat{\theta} = 0$, given a fixed

profile $(\boldsymbol{\pi}^{*-i}, \boldsymbol{\rho}^*)$, it plays the strategy that maximizes its expected payoff based on its belief, and the payoff is defined as $V^{(\pi^i.\boldsymbol{\pi}^{*-i}), \rho^{*v,\hat{\theta}^v}}$ for the type corresponding to $\hat{\theta}^v$.

Moreover, a PBE minimizes the regret in $\mathcal{M}_B$ for the given adversarial profile $\boldsymbol{\rho}^*$. To show that, note that we can write

$$
\begin{aligned}
\arg\min_{\boldsymbol{\pi}} \mathcal{R}(\boldsymbol{\pi}) &= \arg\min_{\boldsymbol{\pi}} \mathbb{E}_{(v,\theta^v)\sim b_0}[\mathcal{R}_{\rho^{*v,\theta^v}}(\boldsymbol{\pi})] \\
&= \arg\min_{\boldsymbol{\pi}} \mathbb{E}_{(v,\theta^v)\sim b_0}[\max_{\boldsymbol{\pi}'}(V^{\boldsymbol{\pi}',\rho^{*v,\theta^v}}) - V^{\boldsymbol{\pi},\rho^{*v,\theta^v}}] \\
&= \arg\min_{\boldsymbol{\pi}}[\mathbb{E}_{(v,\theta^v)\sim b_0}\max_{\boldsymbol{\pi}'}(V^{\boldsymbol{\pi}',\rho^{*v,\theta^v}})] - [\mathbb{E}_{(v,\theta^v)\sim b_0}V^{\boldsymbol{\pi},\rho^{*v,\theta^v}}]. \quad (18)
\end{aligned}
$$

Notice that $\max_{\boldsymbol{\pi}'}(V^{\boldsymbol{\pi}',\rho^{*v,\theta^v}})$ is independent of $\boldsymbol{\pi}$, thus the minimizer above is equivalent to

$$
\arg\min_{\boldsymbol{\pi}} -[\mathbb{E}_{(v,\theta^v)\sim b_0}V^{\boldsymbol{\pi},\rho^{*v,\theta^v}}] = \arg\max_{\boldsymbol{\pi}}[\mathbb{E}_{(v,\theta^v)\sim b_0}V^{\boldsymbol{\pi},\rho^{*v,\theta^v}}] \quad (19)
$$

which is satisfied by $\boldsymbol{\pi}^*$ by definition.

### B.2 EQUIVALENT DEC-POMDP FORMULATION

The Bayesian game $\hat{\mathcal{M}}_B$ ca equivalently be formulated as a partially observable stochastic game $\mathcal{G}$ with $N+1$ players, where player $N+1$ denotes the adversary, and with states $\bar{s} = (s, \hat{\theta})$. Each agent $i \in \mathcal{N}$ only observes its own type $\hat{\theta}^i$ (as part of its observation in $\mathcal{G}$), while the adversary has full observability of the types $\hat{\theta}$. The reward function $\bar{R}^i(\bar{s}, \mathbf{a})$ is the same as $R(s, \mathbf{a})$ for $i \in \mathcal{N}$ and is set to $-R(s, \mathbf{a})$ for player $N+1$.

The initial state distribution is is based on $p(\hat{\theta})$, i.e.,

$$
\bar{\mu}(\bar{s}_0 = (s_0, \hat{\theta})) = \mu(s_0)p(\hat{\theta}), \quad (20)
$$

and the state transition probabilities are defined as

$$
\bar{P}((s', \hat{\theta}')|(s, \hat{\theta}), \mathbf{a}) = \begin{cases} P(s'|s, \mathbf{a}), & \text{if } \hat{\theta}' = \hat{\theta} \\ 0, & \text{otherwise} \end{cases}
$$

Additionally, when $\theta^v = k > 0$, the strategy space of player $N+1$ is restricted to policies in $\Pi_z$, with $z = (v, k)$. In this case, player $v$'s actions become ineffective, which can be modeled using a singleton action set. This model can be considered as a partially observable "Adversarial Team Markov Game" proposed by Kalogiannis et al. (2022).

## C PROOFS

### C.1 PROOF OF PROPOSITION 3.1

First we need to argue that $V_{\max}$ and $V_{\min}^v$ are well-defined. Notice that with full observability assumption, the DEC-MDP model is equivalent to an stochastic game, which always has a Nash equilibrium (Fink, 1964). Since the rewards of all players are identical, the Nash equilibrium corresponds to maximizing $V^{\boldsymbol{\pi}}$ over the set of all policies. Thus, $\boldsymbol{\pi}_0$, and accordingly, $V_{\max}$ exist.

Now if we fix the policies of all non-victim agents to $\boldsymbol{\pi}_0^{-v}$, then the adversary faces a single-agent MDP with the reward function and the state transition probability defined as follows:

$$\bar{R}(s, a^v) = -\sum_{\mathbf{a}^{-v}} R(s, (a^v, \mathbf{a}^{-v}))\boldsymbol{\pi}_0^{-v}(\mathbf{a}^{-v}|s) \tag{21}$$

$$\bar{P}(s'|s, a^v) = \sum_{\mathbf{a}^{-v}} P(s'|s, (a^v, \mathbf{a}^{-v}))\boldsymbol{\pi}_0^{-v}(\mathbf{a}^{-v}|s) \tag{22}$$

Such MDP always has an optimal solution (Puterman, 1994), thus $V_{\min}^v$ also exists.

Now, we show that all sets $\Pi_z$ are non-empty. For a given victim $v$, consider $\bar{\rho}^v \in \arg\min_{\rho^v} V^{\boldsymbol{\pi}_0, \rho_v}$. Define $\rho_\alpha^v = \alpha\bar{\rho}^v + (1-\alpha)\pi_0^v$ for $\alpha \in [0, 1]$. Fix $\boldsymbol{\pi}_0^{-v}$ as the policy of non-victim agents. For any policy, $\pi^v$, the Bellman equation in matrix form is

$$\boldsymbol{v}^{\pi^v} = (\boldsymbol{I} - \gamma\boldsymbol{P}^{\pi^v})^{-1}\boldsymbol{r}^{\pi^v}, \tag{23}$$

where $\boldsymbol{v}^{\pi^v}$ is the vectorized state value function, $\boldsymbol{P}^{\pi^v}$ is a matrix with elements $\boldsymbol{P}_{s's}^{\pi^v} = \sum_{a^v} \bar{P}(s'|s, a^v)\pi^v(a^v|s)$, $\boldsymbol{r}^{\pi^v}$ is a vector with elements $\boldsymbol{r}_s^{\pi^v} = -\sum_{a^v} \bar{R}(s, a^v)\pi^v(a^v|s)$, and $\bar{P}$ and $\bar{R}$ are as defined above. Accordingly, we have $V^{\boldsymbol{\pi}_0, \pi^v} = \boldsymbol{\mu}^T \boldsymbol{v}^{\pi^v}$.

For $\rho_\alpha^v$, it is easy to verify that

$$\boldsymbol{P}^{\rho_\alpha^v} = \alpha\boldsymbol{P}^{\bar{\rho}^v} + (1-\alpha)\boldsymbol{P}^{\pi_0^v} = \alpha(\boldsymbol{P}^{\bar{\rho}^v} - \boldsymbol{P}^{\pi_0^v}) + \boldsymbol{P}^{\pi_0^v} \tag{24}$$

$$\boldsymbol{r}^{\rho_\alpha^v} = \alpha\boldsymbol{r}^{\bar{\rho}^v} + (1-\alpha)\boldsymbol{r}^{\pi_0^v} = \alpha(\boldsymbol{r}^{\bar{\rho}^v} - \boldsymbol{r}^{\pi_0^v}) + \boldsymbol{r}^{\pi_0^v}. \tag{25}$$

Thus, we have

$$V^{\boldsymbol{\pi}_0, \rho_\alpha^v} = \boldsymbol{\mu}^T(\boldsymbol{I} - \gamma\boldsymbol{P}^{\pi_0^v} - \gamma\alpha(\boldsymbol{P}^{\bar{\rho}^v} - \boldsymbol{P}^{\pi_0^v}))^{-1}(\alpha(\boldsymbol{r}^{\bar{\rho}^v} - \boldsymbol{r}^{\pi_0^v}) + \boldsymbol{r}^{\pi_0^v}). \tag{26}$$

Note that, since $\boldsymbol{P}^{\bar{\rho}^v}$ and $\boldsymbol{P}^{\pi_0^v}$ are both row-stochastic matrices, $\boldsymbol{P}^{\rho_\alpha^v}$ is also a row-stochastic matrix and thus, the matrix inverse in (26) always exists, and the result is a continuous function of $\alpha$. Accordingly, $V^{\boldsymbol{\pi}_0, \rho_\alpha^v}$ in (26) is a continuous function of $\alpha$. When $\alpha$ is 0 and 1, $V^{\boldsymbol{\pi}_0, \rho_\alpha^v}$ equals $V_{\max}$ and $V_{\min}^v$, respectively. Thus, as $\alpha$ sweeps between 0 and 1, $V^{\boldsymbol{\pi}_0, \rho_\alpha^v}$ sweeps between $V_{\max}$ and $V_{\min}^v$ continuously. Thus, $\Pi_z$ for all $z \in \mathcal{Z}$ is non-empty.

### C.2 PROOF OF PROPOSITION 3.2

For notational simplicity, we omit the superscript $\theta$. Note that when $\boldsymbol{\pi}_0^{-v}$ is fixed, under the full observability assumption, the DEC-MDP can be viewed as an MDP for agent $v$. Thus, the performance difference lemma (Agarwal et al., 2021) implies that

$$V^{\boldsymbol{\pi}_0, \rho_1^v}(s_0) - V^{\boldsymbol{\pi}_0, \rho_2^v}(s_0) = \mathbb{E}_{s \sim d_{s_0}^{\boldsymbol{\pi}, \rho_1}} \mathbb{E}_{a^v \sim \rho_1(.|s)}\left[A^{\boldsymbol{\pi}_0, \rho_2^v}(s, a^v)\right]. \tag{27}$$

Taking the expectation with respect to $s_0$, we get

$$V^{\boldsymbol{\pi}_0, \rho_1^v} - V^{\boldsymbol{\pi}_0, \rho_2^v} = \mathbb{E}_{s \sim d_\mu^{\boldsymbol{\pi}, \rho_1}} \mathbb{E}_{a^v \sim \rho_1(.|s)}\left[A^{\boldsymbol{\pi}_0, \rho_2^v}(s, a^v)\right]. \tag{28}$$

By taking the absolute value of both sides and applying the Jensen inequality, we can write

$$|V^{\boldsymbol{\pi}_0, \rho_1^v} - V^{\boldsymbol{\pi}_0, \rho_2^v}| \leq \mathbb{E}_{s \sim d_\mu^{\boldsymbol{\pi}, \rho_1}} \left|\mathbb{E}_{a^v \sim \rho_1(.|s)}\left[A^{\boldsymbol{\pi}_0, \rho_2^v}(s, a^v)\right]\right|. \tag{29}$$

Now, we can write

$$\left| \mathbb{E}_{a^v \sim \rho_1(.|s)} \left[ A^{\boldsymbol{\pi}_0, \rho_2^v}(s, a^v) \right] \right| = \left| \sum_{a^v} \rho_1(a^v|s) \left[ Q^{\boldsymbol{\pi}_0, \rho_2^v}(s, a^v) - V^{\boldsymbol{\pi}_0, \rho_2^v}(s) \right] \right|$$

$$= \left| \sum_{a^v} \rho_1(a^v|s) Q^{\boldsymbol{\pi}_0, \rho_2^v}(s, a^v) - \mathbb{E}_{a^v \sim \rho_2(.|s)} \left[ Q^{\boldsymbol{\pi}_0, \rho_2^v}(s, a^v) \right] \right|$$

$$= \left| \sum_{a^v} \left[ \rho_1(a^v|s) - \rho_2(a^v|s) \right] Q^{\boldsymbol{\pi}_0, \rho_2^v}(s, a^v) \right|$$

$$\leq \sum_{a^v} |\rho_1(a^v|s) - \rho_2(a^v|s)| \, Q^{\boldsymbol{\pi}_0, \rho_2^v}(s, a^v). \tag{30}$$

Note that the reward function is bounded in $[-1, 1]$, thus the Q-function is bounded by $\sum_{t=0}^{\infty} 1\gamma^t = \frac{1}{1-\gamma}$. Thus, we conclude that

$$\left| \mathbb{E}_{a^v \sim \rho_1(.|s)} \left[ A^{\boldsymbol{\pi}_0, \rho_2^v}(s, a^v) \right] \right| \leq \frac{||\rho_1(s) - \rho_2(s)||_1}{1 - \gamma}, \tag{31}$$

and accordingly,

$$|V^{\boldsymbol{\pi}_0, \rho_1^v} - V^{\boldsymbol{\pi}_0, \rho_2^v}| \leq \frac{1}{1 - \gamma} \mathbb{E}_{s \sim d_\mu^{\boldsymbol{\pi}, \rho_1}} ||\rho_1(s) - \rho_2(s)||_1. \tag{32}$$

On the other hand, the Pinsker inequality (Csiszár & Körner, 2011) implies that

$$||\rho_1(s) - \rho_2(s)||_1 \leq \sqrt{2 D_{\mathrm{KL}}(\rho_1^v(s) || \rho_2^v(s))}, \quad \forall s \in \mathcal{S}. \tag{33}$$

Thus, by taking the expectation and applying Jensen's inequality to the concave square-root function, one can obtain

$$|V^{\boldsymbol{\pi}_0, \rho_1^v} - V^{\boldsymbol{\pi}_0, \rho_2^v}| \leq \frac{1}{(1 - \gamma)} \sqrt{2 \mathbb{E}_{s \sim d_\mu^{\boldsymbol{\pi}, \rho_1}} \left[ D_{\mathrm{KL}}(\rho_1^v(s) || \rho_2^v(s)) \right]}, \tag{34}$$

and therefore,

$$\frac{(1 - \gamma)^2}{2} |V^{\boldsymbol{\pi}_0, \rho_1^v} - V^{\boldsymbol{\pi}_0, \rho_2^v}|^2 \leq \mathbb{E}_{s \sim d_\mu^{\boldsymbol{\pi}, \rho_1}} \left[ D_{\mathrm{KL}}(\rho_1^v(s) || \rho_2^v(s)) \right]. \tag{35}$$

### C.3 Proof of Proposition 3.3

Note that by definition, we have $\rho^{*v,k} \in \Pi_z$ for $z = (v, k)$. Accordingly,

$$\frac{k - 1}{K} \leq \eta_{\rho^{*v,k}} \leq \frac{k}{K}. \tag{36}$$

We can write $V^{\boldsymbol{\pi}_0, \rho^{*v,k}} = V_{\max} - \eta_{\rho^{*v,k}}(V_{\max} - V_{\min}^v)$. Thus, applying Proposition 3.2 leads to

$$\mathbb{E}_{s \sim d_\mu^{\boldsymbol{\pi}_0, \rho^{*v,i}}} \left[ D_{\mathrm{KL}}(\rho^{*v,i}(s) || \rho^{*v,j}(s)) \right] \geq \frac{(1 - \gamma)^2}{2} (V_{\max} - V_{\min}^v)^2 |\eta_{\rho^{*v,i}} - \eta_{\rho^{*v,j}}|^2. \tag{37}$$

Therefore, the diversity can be lower bounded as

$$\mathrm{Div}(\{\rho^{*v,k}\}_{k=1}^K) \geq \frac{(1 - \gamma)^2}{2K(K - 1)} (V_{\max} - V_{\min}^v)^2 \sum_{i \neq j} |\eta_{\rho^{*v,i}} - \eta_{\rho^{*v,j}}|^2. \tag{38}$$

Thus, it is sufficient to find the minimum of $S_K = \sum_{i \neq j} |\eta_{\rho^{*v,i}} - \eta_{\rho^{*v,j}}|^2$ conditioned on (36). For the ease of notation we denote $\eta_{\rho^{*v,i}}$ by $\eta_i$ in the rest of the proof. We claim the following: If $K$ is even, the minimum of $S_K$ is attained when $\eta_i = \frac{i}{K}$ for $i \leq \frac{K}{2}$ and $\eta_i = \frac{i-1}{K}$ for $i \geq \frac{K}{2} + 1$. If $K$ is odd, the minimum of $S_K$ is attained when $\eta_i = \frac{i}{K}$ for $i \leq \frac{K-1}{2}$, $\eta_i = \frac{i-1}{K}$ for $i \geq \frac{K+1}{2} + 1$, and $\eta_{\frac{K+1}{2}} = \frac{1}{2}$. Moreover, in both cases the minimum is $\frac{(K-1)(K-2)}{6}$.

We prove this claim by induction. We prove that if it holds for some $K$, it will hold for $K + 2$ as well. The bases for our induction are $K = 2$, and $K = 3$. The case with $K = 2$ is trivial. For $K = 3$, we can write

$$S_3 = 2((\eta_3 - \eta_1)^2 + (\eta_2 - \eta_1)^2 + (\eta_3 - \eta_2)^2) \geq 2((\frac{1}{3})^2 + (\eta_2 - \frac{1}{3})^2 + (\frac{2}{3} - \eta_2)^2). \tag{39}$$

The minimizer of the right hand side of (39) is $\eta_2 = \frac{1}{2}$, and the left and right hand sides are equal when $\eta_1 = \frac{1}{3}$ and $\eta_3 = \frac{2}{3}$. Thus, these values correspond to the minimum of $S_3$, which is $\frac{1}{3}$. This proves our claim for $K = 3$.

Now we prove that if thee claim holds for $K = K_1$, it holds for $K = K_1 + 2$. We can write

$$S_{K_1+2} = 2\left((\eta_{K_1+2} - \eta_1)^2 + \sum_{2 \leq i < j \leq K_1+1} (\eta_i - \eta_j)^2 + \sum_{k=2}^{K_1+1} (\eta_k - \eta_1)^2 + \sum_{k=2}^{K_1+1} (\eta_{K_1+2} - \eta_k)^2\right). \tag{40}$$

Note that $S_K$ is invariant with respect to any constant shift of all elements, accordingly the minimum of $\sum_{2 \leq i < j \leq K_1+1} (\eta_i - \eta_j)^2$ is achieved by the same relative configuration as the minimizer of $S_{K_1}$. Based on the assumption of induction, the minimum of $S_{K_1}$ is $\frac{(K_1-1)(K_1-2)}{6}$. Then, according to the scaling from $K_1$ to $K_1 + 2$, the minimum of $2\sum_{2 \leq i < j \leq K_1+1} (\eta_i - \eta_j)^2$ is $S' = (\frac{K_1}{K_1+2})^2 \frac{(K_1-1)(K_1-2)}{6}$. Accordingly, given (36) we have:

$$S_{K_1+2} \geq 2\left((\frac{K_1}{K_1+2})^2 + S'/2 + \sum_{k=2}^{K_1+1} (\eta_k - \frac{1}{K_1+2})^2 + \sum_{k=2}^{K_1+1} (\frac{K_1+1}{K_1+2} - \eta_k)^2\right). \tag{41}$$

The equality holds when $\eta_1 = \frac{1}{K_1+2}$, $\eta_{K_1+2} = \frac{K_1+1}{K_1+2}$, and the rest of $\eta_i$'s are as specified in our claim (due to the assumption of induction and the fact that $K$ and $K + 2$ have the same parity). If $K_1$ is even, the first summation in (41) in this configuration is equal to

$$\sum_{k=1}^{K_1/2} (\frac{k}{K_1+2})^2 + \sum_{k=K_1/2}^{K_1-1} (\frac{k}{K_1+2})^2 = \frac{1}{(K_1+2)^2}((\frac{K_1}{2})^2 + \sum_{k=1}^{K_1-1} k^2). \tag{42}$$

The second summation is also equal to the same value. Thus, the value of $S_{K_1+2}$ in this configuration becomes

$$\begin{aligned}
S_{K_1+2} &= 2\frac{1}{(K_1+2)^2}\left(K_1^2 + \frac{(K_1-1)(K_1-2)}{12} + 2[(\frac{K}{2})^2 + \sum_{k=1}^{K} k^2]\right) \\
&= 2\frac{1}{(K_1+2)^2}\left(\frac{3}{2}K_1^2 + \frac{(K_1-1)(K_1-2)}{12} + \frac{K_1(K_1-1)(2K_1-1)}{3}\right) \\
&= 2\frac{1}{(K_1+2)^2}(\frac{K_1(K_1^3 + 5K_1^2 + 8K_1 + 4)}{12}) = 2\frac{1}{(K_1+2)^2}(\frac{K_1(K_1+1)(K_1+2)^2}{12}).
\end{aligned} \tag{43}$$

which is simplified to $\frac{K_1(K_1+1)}{6}$. Similar calculations for odd $K_1$ leads to the same result. This is the same expression as we claimed for $K = K_1 + 2$, which concludes the proof.

### C.4 PROOF OF PROPOSITION 3.4

We have

$$\mathcal{R}_{\hat{\rho}^v}(\pi^*) = (\max_{\pi} V^{\pi, \hat{\rho}^v}) - V^{\pi_z^*, \hat{\rho}^v}. \tag{44}$$

Note that $\max_{\pi} V^{\pi, \hat{\rho}_z^v} \leq \max_{\pi} V^{\pi} = V_{\max}$. Also, we can write

$$V^{\pi_z^*, \hat{\rho}^v} \geq \min_{\rho^v \in \Pi_z} V^{\pi_z^*, \rho^v}. \tag{45}$$

However, based on the definition of $\pi_z^*$, we know that $\min_{\rho^v \in \Pi_z} V^{\pi_z^*, \rho^v} = \max_{\pi} \min_{\rho^v \in \Pi_z} V^{\pi, \rho^v}$. Thus, it follows that

$$V^{\pi_z^*, \hat{\rho}^v} \geq \min_{\rho^v \in \Pi_z} V^{\pi_0, \rho^v} = V_{\max} - \frac{k}{K}(V_{\max} - V_{\min}^v). \tag{46}$$

Thus, we conclude that $\mathcal{R}_{\hat{\rho}^v}(\pi^*) \leq \frac{k}{K}(V_{\max} - V_{\min}^v)$.

## C.5 PROOF OF PROPOSITION 4.1

The policy gradient theorem (Sutton & Barto, 2018) implies that

$$\nabla_\psi \mathbb{E}_{s\sim\mu}[V_{(j)}^{\rho_\psi}(s)] = \mathbb{E}_{s\sim d_{(j)}, \, a\sim\rho_\psi(.|s)}[\nabla_\psi \log \rho_\psi(a|s) A_{(j)}^{\rho_\psi}(s,a)], \quad j = 0, 1. \tag{47}$$

Thus, we can write

$$\nabla_\psi \log(\mathbb{E}_{s\sim\mu}[V_{(0)}^{\rho_\psi}(s)] - l) = \frac{\nabla_\psi \mathbb{E}_{s\sim\mu}[V_{(0)}^{\rho_\psi}(s)]}{\mathbb{E}_{s\sim\mu}[V_{(0)}^{\rho_\psi}(s)] - l} = \frac{\mathbb{E}_{s\sim d_{(0)}, \, a\sim\rho_\psi(.|s)}[\nabla_\psi \log \rho_\psi(a|s) A_{(0)}^{\rho_\psi}(s,a)]}{\mathbb{E}_{s\sim\mu}[V_{(0)}^{\rho_\psi}(s)] - l}$$

$$\nabla_\psi \log(h - \mathbb{E}_{s\sim\mu}[V_{(0)}^{\rho_\psi}(s)]) = \frac{-\nabla_\psi \mathbb{E}_{s\sim\mu}[V_{(0)}^{\rho_\psi}(s)]}{h - \mathbb{E}_{s\sim\mu}[V_{(0)}^{\rho_\psi}(s)]} = \frac{-\mathbb{E}_{s\sim d_{(0)}, a\sim\rho_\psi(.|s)}[\nabla_\psi \log \rho_\psi(a|s) A_{(0)}^{\rho_\psi}(s,a)]}{h - \mathbb{E}_{s\sim\mu}[V_{(0)}^{\rho_\psi}(s)]}.$$

This proves the result.

## C.6 PROOF OF PROPOSITION 4.2

First let us establish the following preliminaries.

**Lemma C.1.** *let $G$ and $H$ be the upper bounds on $||\nabla_\psi \log \rho_\psi(a|s)||$ and $||\nabla_\psi^2 \log \rho_\psi(a|s)||$, respectively. Then,*

- *The variance of $\hat{V}_{(0)}^\psi$ is bounded by $\sigma^2(M) = \frac{1}{M(1-\gamma)^2}$.*

- *The variance of the gradient estimates of the value function, i.e., $\frac{1}{1-\gamma}\hat{\nabla}_{(j)}^\psi$, is bounded by $\bar{\sigma}^2(M) = \frac{4G^2}{M(1-\gamma)^4}$.*

- *$V_{(j)}^{\rho_\psi}$ is Lipschitz continuous with respect to $\psi$ with constant $L = \frac{2G}{(1-\gamma)^2}$. It is also B-smooth with smoothness constant $B = \frac{1}{(1-\gamma)^2}\left(\frac{1+\gamma}{1-\gamma}G^2 + H\right)$.*

*Proof.* First notice that if $\sigma$ is the variance bound per one sample (one episode), then $\sigma(M) = \frac{1}{\sqrt{M}}\sigma$. To bound $\sigma$, note that as $||r_t|| \le 1$, $||\sum_{t=0}^\infty \gamma^t r_t|| \le \frac{1}{1-\gamma}$, so $\hat{V}_{(0)}^\psi \in [-\frac{1}{1-\gamma}, \frac{1}{1-\gamma}]$, and $\sigma$ cannot be larger than $\frac{1}{1-\gamma}$.

Regarding the variance of the gradient estimation, note that for any $s_t, a_t$:

$$\frac{1}{1-\gamma}||\nabla_\psi \log \rho^\psi(a_t|s_t) A^{\rho_\psi}(s_t, a_t)|| \le \frac{1}{(1-\gamma)}G \max_{s,a}|A^{\rho_\psi}(s,a)|. \tag{48}$$

Moreover, we have

$$|A^{\rho_\psi}(s,a)| = |Q^{\rho_\psi}(s,a) - V^{\rho_\psi}(s)| \le |Q^{\rho_\psi}(s,a)| + |V^{\rho_\psi}(s)| \le \frac{2}{1-\gamma}. \tag{49}$$

Thus, each per sample (per episode) estimate of the gradient is norm-bounded by $\frac{2G}{(1-\gamma)^2}$, which implies that

$$\mathbb{E}[||\frac{1}{1-\gamma}\hat{\nabla}_{(j)}^\psi - \nabla_\psi V_{(j)}^{\rho_\psi}||^2] \le \frac{4G^2}{M(1-\gamma)^4}. \tag{50}$$

The bound on the norm range of $\nabla_\psi V^{\rho_\psi}$ will also implies Lipschitz continuity with the same bound $L = \frac{2G}{(1-\gamma)^2}$.

Finally, the smoothness constant is the direct consequence of Lemma 6 in (Papini et al., 2022) by setting the bounds $\mathbb{E}[||\nabla_\psi \log \rho_\psi(a|s)||] \le G$, $\mathbb{E}[||\nabla_\psi \log \rho_\psi(a|s)||^2] \le G^2$, and $\mathbb{E}[||\nabla_\psi^2 \log \rho_\psi(a|s)||] \le H$.

$\square$

In our analysis we use the results of Usmanova et al. (2024), however, given the structure of our problem we are able to derive simpler step-sizes. Moreover, we base our proof on a required error $\epsilon$ on the true gradient $g_\psi$ given a fixed $\lambda$ instead of a fixed required error on the noisy gradients.

To use these results of Usmanova et al. (2024), first we have to confirm that the required assumptions hold. The Lipschtiz continuity and smoothness of $V_{(k)}^{\rho_\psi}$ are already established using the above lemma, and assumption 3 in Proposition 4.2 ensures a feasible starting point. Moreover, since the gradient of our constraints differ only on their signs, the extended Mangasarian-Fromovitz constraint qualification (MFCQ) requirements proposed by Usmanova et al. (2024) is equivalent to the requirement that "there are positive constants $\zeta$ and $q$, such that $||\nabla_\psi V_{(0)}^\psi|| \geq q$ when $h - \zeta \leq V_{(0)}^{\rho_\psi} \leq h$ or $l \leq V_{(0)}^{\rho_\psi} \leq l + \zeta$". Assumption 4 in Proposition 4.2 guarantees this condition. This is because $\nabla_\psi V_{(0)}^{\rho_\psi}$ is differentiable in $[l, l + \zeta]$ and $[h - \zeta, h]$ and hence $||\nabla_\psi V_{(0)}^{\rho_\psi}||$ is continuous over these sets. Accordingly, the Extreme Value Theorem guarantees the existence of a minimum in each of these sets. We define $q$ as the minimum among these two minimums, and it is indeed positive.

Let us define $c = 0.5(\frac{q}{20L})^2$ and

$$C = \frac{c}{2L^2(1 + \frac{2}{c}) \max\{4 + \frac{5Bc\lambda}{L^2}, 1 + \sqrt{\frac{Bc\lambda}{4L^2}},\}}, \tag{51}$$

where $B$ and $L$ are as defined in Lemma C.1. We denote by $F_\lambda$ the log-barrier regularized objective function in (9). Now define

$$N_{iter} = \frac{3(F_\lambda(\psi_0) + \frac{1}{1-\gamma} + 2\lambda \log(h - l))}{32\epsilon^2 C\lambda}, \tag{52}$$

and let $\hat{\delta} = \frac{\delta}{2N_{iter}}$.

To find the local smoothness constant, we further require to define $x_n^1 = V_{(0)}^{\rho_{\psi_n}} - l$, $x_n^2 = h - V_{(0)}^{\rho_{\psi_n}}$, $\bar{x}_n^1 = \hat{V}_{(0)}^{\psi_n} - l$, and $\bar{x}_n^2 = h - \hat{V}_{(0)}^{\psi_n}$. Moreover, assume $\underline{x}_n^j = \bar{x}_n^j - \sigma(M)\sqrt{\ln \frac{1}{\delta}}$.

Now we are ready to introduce the adaptive step-size and the local smoothness constant $F_\lambda$. Let $\hat{B}_2 = B + 10B\lambda(\frac{1}{\underline{x}_n^1} + \frac{1}{\underline{x}_n^2}) + 8L^2\lambda(\frac{1}{(\underline{x}_n^1)^2} + \frac{1}{(\underline{x}_n^2)^2})$,. Moreover, let $D_n \triangleq \min\{\frac{x_n^1}{2L + \sqrt{B\underline{x}_n^1}}, \frac{x_n^2}{2L + \sqrt{B\underline{x}_n^2}}\}$. We define the step size

$$\alpha_n = \min\left\{\frac{D_n}{||g_{\hat{\psi}_n}||}, \frac{1}{\hat{B}_2}\right\}. \tag{53}$$

**Lemma C.2.** *The function $F_\lambda$ is locally smooth and its smoothness constant is less $\hat{B}_2$ with probability at least $1 - \hat{\delta}$.*

*Proof.* Let $y_n = \langle \nabla_\psi V_{(0)}^{\rho_{\psi_n}}, \frac{g_{\psi_n}}{||g_{\psi_n}||} \rangle$. Lemma 2 in Usmanova et al. (2024) implies that $F_\lambda$ is locally smooth with constant

$$B_2(\psi_n) = B + 10\lambda(\frac{B}{x_n^1} + \frac{B}{x_n^2}) + 8\lambda((\frac{y_n}{x_n^1})^2 + (\frac{y_n}{x_n^2})^2) \tag{54}$$

as long as $\alpha_n \leq \min\{\frac{x_n^1}{2y_n + \sqrt{x_n^1 B}}, \frac{x_n^2}{2y_n + \sqrt{x_n^2 B}}\}$, and $x_{n+1}^j \geq \frac{x_n^j}{2}$ for $j = 1, 2$.

Now, note that we can consider $\bar{x}_n^j$ is a lower bound on $x_n^j$ with probability $1 - \hat{\delta}$, and we have

$$y_n = \langle \nabla_\psi V_{(0)}^{\rho_{\psi_n}}, \frac{g_{\psi_n}}{||g_{\psi_n}||} \rangle = ||\nabla_\psi V_{(0)}^{\rho_{\psi_n}}|| \langle \frac{\nabla_\psi V_{(0)}^{\rho_{\psi_n}}}{||\nabla_\psi V_{(0)}^{\rho_{\psi_n}}||}, \frac{g_{\psi_n}}{||g_{\psi_n}||} \rangle \leq ||\nabla_\psi V_{(0)}^{\rho_{\psi_n}}|| \leq L. \tag{55}$$

Thus, the step size $\alpha_n$ satisfies the requirement with probability at least $1 - \hat{\delta}$. Moreover, Lemma 3 in Usmanova et al. (2024) implies that $x_{n+1}^j \geq \frac{x_n^j}{2}$, which concludes the proof. $\square$

Based on the smoothness of $F_\lambda$, for any $n$ with probability $1 - \hat{\delta}$ we have

$$F_\lambda(\psi_n) - F_\lambda(\psi_{n+1}) \geq \alpha_n \langle g_{\psi_n}, \hat{g}_{\psi_n} \rangle - \frac{1}{2} \hat{B}_2 \alpha_n^2 ||g_{\hat{\psi}_n}||^2 \geq \frac{1}{2} \alpha_n ||g_{\hat{\psi}_n}||^2 - \alpha_n ||g_{\hat{\psi}_n}|| ||g_{\psi_n} - \hat{g}_{\psi_n}||.$$
(56)

The last inequality holds because of the selection of the step sizes. which implies that $\alpha_n \hat{B}_2 \leq 1$ with probability $1 - \hat{\delta}$.

Note that with this selection of step sizes, Theorem 4 in Usmanova et al. (2024) implies the feasibility of all $\psi_n$ for all $n = 1, 2, ..., N_{iter}$ with probability at least $1 - 2N_{iter}\hat{\delta} = 1 - \delta$. Then, by summing up the above inequality for $0 \leq n < N_{iter}$, we obtain

$$N_{iter} \min_n \left[ \alpha_n ||g_{\hat{\psi}_n}|| (\frac{1}{2} ||g_{\hat{\psi}_n}|| - ||g_{\psi_n} - \hat{g}_{\psi_n}||) \right] \leq \sum_{n=0}^{N_{iter}-1} \alpha_n ||g_{\hat{\psi}_n}|| (\frac{1}{2} ||g_{\hat{\psi}_n}|| - ||g_{\psi_n} - \hat{g}_{\psi_n}||)$$
$$\leq F_\lambda(\psi_0) - \min_\psi F_\lambda(\psi)$$
$$\leq F_\lambda(\psi_0) + \frac{1}{1 - \gamma} + 2\lambda \log(h - l) \quad (57)$$

w.p $1 - \delta$. Accordingly, given the definition of $N_{iter}$, we obtain

$$\min_n \left[ \alpha_n ||g_{\hat{\psi}_n}|| (\frac{1}{2} ||g_{\hat{\psi}_n}|| - ||g_{\psi_n} - \hat{g}_{\psi_n}||) \right] \leq \frac{3}{32} \epsilon^2 C\lambda. \quad (58)$$

To bound $||g_{\psi_n} - \hat{g}_{\psi_n}||$, assume that $M$ is large enough such that

$$\sigma(M) \leq \frac{\epsilon}{20\lambda L \sqrt{\log \frac{1}{\hat{\delta}}}} \min\{(\underline{x}_n^1)^2, (\underline{x}_n^2)^2\},$$

$$\hat{\sigma}(M) \leq \frac{\epsilon}{20\sqrt{\log \frac{1}{\hat{\delta}}}} \min\left\{1, \frac{x_n^1 L}{\lambda}, \frac{x_n^2 L}{\lambda}\right\} \quad (59)$$

Then, Lemma 1 in Usmanova et al. (2024) implies that w.p. at least $1 - \hat{\delta}$ we have

$$||g_{\psi_n} - \hat{g}_{\psi_n}|| \leq \hat{\sigma}(M) \sqrt{\log \frac{1}{\hat{\delta}}} + \lambda \hat{\sigma}(M) \sqrt{\log \frac{1}{\hat{\delta}}} (\frac{1}{\bar{x}_n^1} + \frac{1}{\bar{x}_n^2}) + \lambda L \sigma(M) \sqrt{\log \frac{1}{\hat{\delta}}} (\frac{1}{x_n^1 \bar{x}_n^1} + \frac{1}{x_n^2 \bar{x}_n^2})$$
$$\leq \frac{5\epsilon}{20} = \frac{\epsilon}{4}. \quad (60)$$

Thus, w.p. $1 - \delta$, $||g_{\psi_n} - \hat{g}_{\psi_n}|| \leq \frac{\epsilon}{4}$ for all $0 \leq n < N_{iter}$. On the other hand, by Lemma 5 in Usmanova et al. (2024) we know $\alpha_n \geq C\lambda$ for all $0 \leq n < N_{iter}$ w.p. $1 - \delta$. Thus we can write

$$\frac{3}{32} \epsilon^2 C\lambda \geq \min_n \left[ \alpha_n ||g_{\hat{\psi}_n}|| (\frac{1}{2} ||g_{\hat{\psi}_n}|| - ||g_{\psi_n} - \hat{g}_{\psi_n}||) \right] \geq C\lambda (\frac{1}{2} g^2 - g \frac{\epsilon}{4}), \quad (61)$$

where $g \triangleq \min_n ||\hat{g}_{\psi_n}||$. Therefore, we have $\frac{1}{2} g^2 - g \frac{\epsilon}{4} \leq \frac{3}{32} \epsilon^2$. Solving for $g$ (and given its positivity) we obtain

$$g \leq \frac{\epsilon}{4} + \sqrt{\frac{\epsilon^2}{16} + 3\frac{\epsilon^2}{16}} = \frac{3\epsilon}{4}. \quad (62)$$

Finally, we obtain that w.p $1 - \delta$:

$$\min_n ||g_{\psi_n}|| \leq \min_n (||\hat{g}_{\psi_n}|| + ||g_{\psi_n} - \hat{g}_{\psi_n}||) \leq \frac{\epsilon}{4} + \frac{3\epsilon}{4} = \epsilon, \quad (63)$$

which concludes the proof for the first part. Moreover, approaching the solution to a KKT point of the constrained problem is the direct consequence of Lemma 7 in Usmanova et al. (2024) when $\lambda \to 0$.

# D  ADDITIONAL RESULTS

## D.1  LEARNING CURVES OF THE BASELINES

Figure 3 shows the learning curves of the baselines in their training phase against their own adversarial policy.

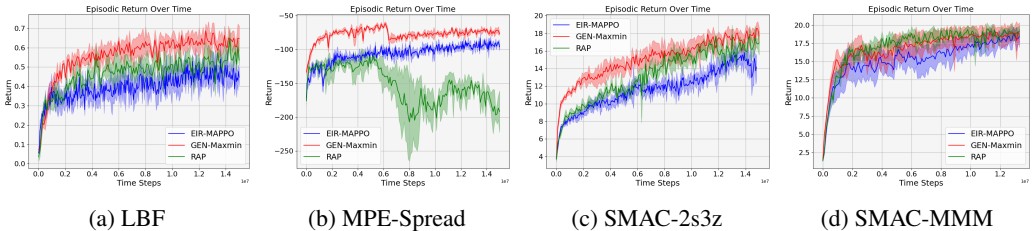

(a) LBF          (b) MPE-Spread          (c) SMAC-2s3z          (d) SMAC-MMM

Figure 3: Average episodic return of the baselines during adversarial learning.

## D.2  EVALUATION OF ROMANCE

ROMANCE (Yuan et al., 2023) is a c-MARL framework designed for adversaries with a limited budget of action manipulations. Thus its comparison in our setting is not fair, however, for the sake of completeness we report its performance against attacks trained in BATPAL and also dynamic adversaries in SMAC-2s3z. We used the already trained models in the original implementation.

Table 2: Win rate of ROMANCE against different attacks in SMAC-2s3z

| Attack | Win Rate |
|---|---|
| No Attack | 0.97 |
| Severity 1 | 0.23 |
| Severity 2 | 0.08 |
| Severity 3 | 0.07 |
| Severity 4 | 0.05 |
| ACT | 0.05 |
| DYN-1 | 0.07 |
| DYN-2 | 0.06 |

## D.3  REFERENCE-VALUE EVALUATION

Figure 4 shows the normalized initial state value function of the different attacks trained for BATPAL against the reference policy $\pi_0$ in all environments. This figure shows how the type of the attacks changed during the training.

## D.4  COMPARISON OF DIFFERENT VALUES OF $K$

Table 3 shows the evaluation of policies learned by BATPAL with different number of severity types $K$. It can be observed that, as expected, increasing the number of adversarial types generally enhances the robustness of the c-MARL policy. However, each additional severity type requires the introduction of an additional network, which increases the overall training time. Nonetheless, the results indicate that even with a relatively small number of severity levels (e.g., $K = 4$), satisfactory performance can be achieved across most scenarios

## D.5  MULTIPLE VICTIMS

BATPAL can be adapted for robustness to attacks against multiple victims. Consider a powerful adversary that launches a coordinated attack on $N_v$ victim agents $v_1, v_2, ..., v_{N_v}$. We can group the

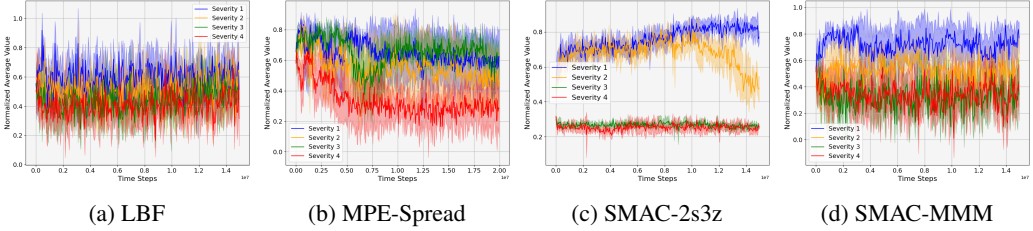

| (a) LBF | (b) MPE-Spread | (c) SMAC-2s3z | (d) SMAC-MMM |

Figure 4: Normalized average initial state value of the BATPAL attacks against the reference policy.

Table 3: BATPAL performance for different values of $K$ in four environments.

| Environment | Attack | $K = 3$ | $K = 4$ | $K = 5$ |
|---|---|---|---|---|
| MPE-Spread | No attack | 0.99 | 1.0 | 1.0 |
| | A-EIR-MAPPO | 0.72 | 0.64 | 0.73 |
| | ACT | 0.79 | 0.81 | 0.89 |
| | DYN-1 | 0.64 | 0.67 | 0.71 |
| | DYN-2 | 0.70 | 0.72 | 0.70 |
| LBF | No attack | 1.0 | 1.0 | 1.01 |
| | A-EIR-MAPPO | 0.20 | 0.26 | 0.38 |
| | ACT | 0.18 | 0.35 | 0.35 |
| | DYN-1 | 0.44 | 0.52 | 0.53 |
| | DYN-2 | 0.52 | 0.58 | 0.60 |
| SMAC-2s3z | No attack | 0.94 | 0.98 | 0.98 |
| | A-EIR-MAPPO | 0.1 | 0.38 | 0.29 |
| | ACT | 0.16 | 0.72 | 0.79 |
| | DYN-1 | 0.23 | 0.52 | 0.53 |
| | DYN-2 | 0.55 | 0.71 | 0.76 |
| SMAC-MMM | No attack | 1 | 1 | 1 |
| | A-EIR-MAPPO | 0.83 | 0.85 | 0.87 |
| | ACT | 0.87 | 0.89 | 0.88 |
| | DYN-1 | 0.83 | 0.82 | 0.88 |
| | DYN-2 | 0.89 | 0.92 | 0.92 |

victim agents into a single "compound" agent whose action and observation spaces correspond to the joint actions and observations of the victim agents. Doing so results in a system with $N - N_v$ benign agents, one adversary, and corresponding severity scores. Our analysis applies in this setting, and BATPAL can be used for training. The main shortcoming of this approach is the combinatorial explosion of the sets of victim agents with increasing $N_v$, as the prior distribution would have to be defined on all possible combinations of $N_v$ agents, which would also have to be explored during training.

Importantly, this formulation eliminates the need to define a separate severity level for each victim agent; instead, the severity is defined for the entire group as a whole. Moreover, in principle, it can provide robustness to attacks against less than $N_v$ agents, since, for example, an attack on $N_v - 1$ agents can be viewed as an attack on $N_v$ agents with lower severity.

We evaluated this approach by training a model with $N_v = 3$ victims for SMAC-MMM, and testing it when two or three agents are manipulated using different attacks. Table 4 presents the results of this evaluation. Since EIR-MAPPO is designed for a single victim, we include comparisons only with the other two baselines. The results show that BATPAL is superior to the baselines in the multiple-victim setting as well. However, for a severe attack such as ACT, when three agents are attacked, the task becomes so challenging that none of the methods is able to produce a good policy.

Table 4: Win rate of different methods in SMAC-MMM against attacks to multiple victims.

|            | Attack     | BATPAL | Gen-Maxmin | RAP  |
|------------|------------|--------|------------|------|
|            | Severity 1 | 0.64   | 0.68       | 0.40 |
|            | DYN 1      | 0.44   | 0.16       | 0.27 |
| 2 Victims  | ACT        | 0.32   | 0.06       | 0.24 |
|            | Severity 1 | 0.28   | 0.18       | 0.14 |
|            | DYN 1      | 0.10   | 0.00       | 0.00 |
| 3 Victims  | ACT        | 0.00   | 0.00       | 0.00 |

## D.6 ADDITIONAL ENVIRONMENTS

In addition to the results shown in Figure 2, we provide here results for two additional SMAC scenarios, namely 1c3s5z and 11m. It can be observed that BATPAL remains superior to the other baselines in providing consistent robustness, especially in SMAC-1c3s5z, where the performance gap between BATPAL and the other methods is substantial.

Table 5: Win rate of different methods in SMAC-1c3s5z and SMAC-11m.

| Environment | Method     | No Attack | Severity 1 | Severity 2 | Severity 3 | Severity 4 | A-EIR-MAPPO | A-Gen-Maxmin | A-RAP |
|-------------|------------|-----------|------------|------------|------------|------------|-------------|--------------|-------|
|             | BATPAL     | **1.00**  | **0.87**   | **0.96**   | **0.88**   | **0.95**   | **0.87**    | **0.89**     | 0.92  |
|             | EIR-MAPPO  | 0.90      | 0.53       | 0.43       | 0.52       | 0.52       | 0.56        | 0.41         | 0.47  |
| SMAC-1c3s5z | Gen-Maxmin | 0.90      | 0.45       | 0.35       | 0.43       | 0.42       | 0.38        | 0.79         | 0.37  |
|             | RAP        | 0.98      | 0.64       | 0.85       | **0.88**   | 0.87       | 0.77        | 0.64         | **0.94** |
|             | BATPAL     | 0.96      | 0.82       | **0.86**   | **0.63**   | **0.54**   | 0.29        | 0.69         | **0.44** |
|             | EIR-MAPPO  | 0.96      | 0.86       | 0.79       | 0.51       | 0.45       | **0.51**    | 0.41         | 0.39  |
| SMAC-11m    | Gen-Maxmin | **1.00**  | **0.90**   | 0.83       | 0.52       | 0.33       | 0.23        | **0.82**     | 0.32  |
|             | RAP        | 0.98      | 0.86       | **0.86**   | 0.46       | 0.40       | 0.49        | 0.64         | **0.44** |

# E  IMPLEMENTATION

## E.1  PSEUDO CODE

The pseudo-code of our algorithm is shown in Algorithm 1. One important point is that when updating the policy networks, we sample the type according to the prior distribution and update only the corresponding network. Consequently, at each update, at most one adversarial network is updated.

---

**Algorithm 1** Adversarial Learning in BATPAL

---

**Input Networks:** The reference policy networks $\omega_0^i$, the policy networks $\omega^i$, the critic $\phi_{(1)}$, the reference critic $\phi_{(0)}$, the belief networks $\chi^i$, and the adversarial policies $\psi^z$

1: **Pretraining:**
2: Train the c-MARL team in a non-adversarial environment and obtain $\boldsymbol{\omega}_0$, $V_{\max}$, and $V_{\min}^v$

3: **Adversarial Training:**
4: **for** each iteration **do**
5:     Sample $\hat{\theta} \sim p$, and set $z = (k, v)$ or $z = 0$.
6:     **for** $m = 1, 2, \ldots, M$ **do**         ▷ Storing experiences corresponding to POMDP$_1$
7:         Sample the initial state and observations $s_{0,(1)}, \mathbf{o}_{0,(1)}$
8:         **for** $t = 0, 1, 2, \ldots, T_m - 1$ **do**
9:             Sample non-victim actions $a_{t,(1)}^i \sim \pi_{\omega^i}(.|\tau_{t,(1)}^i, b_t^i)$, where $b_t^i = b_{\chi^i}(\tau_{t,(1)}^i)$
10:           Sample adversary action if $z \neq 0$, $a_t^v \sim \rho_{\psi^z}$
11:           Set the joint action profile $\mathbf{a}_t = (a_t^v, \mathbf{a}_t^{-v})$ if $z \neq 0$
12:           Environment transitions and $s_{t+1,(1)}, r_{t,(1)}$ are obtained
13:           Store the transition history $H_{t,(1)}$
14:         **end for**
15:     **end for**
16:     **if** $z \neq 0$ **then**         ▷ Storing experiences corresponding to POMDP$_0$
17:         **for** $m = 1, 2, \ldots, M$ **do**
18:             Sample the initial state and observations $s_{0,(0)}, \mathbf{o}_{0,(0)}$
19:             **for** $t = 0, 1, 2, \ldots, T_m - 1$ **do**
20:                 Sample non-victim actions $a_{t,(0)}^i \sim \pi_{\omega_0^i}(.|\tau_{t,(0)}^i)$
21:                 Sample adversary action $a_t^v \sim \rho_{\psi^z}$
22:                 Set the joint action profile $\mathbf{a}_t = (a_t^v, \mathbf{a}_t^{-v})$
23:                 Environment transitions and $s_{t+1,(0)}, r_{t,(0)}$ are obtained
24:                 Store the transition history $H_{t,(0)}$
25:             **end for**
26:         **end for**
27:     **end if**
28:     Update the critics $\phi_{(0)}$. $\phi_{(1)}$ using the advantages obtained by $H_{t,(0)}$ and $H_{t,(1)}$
29:     Update $\chi^i$ using true types in $H_{t,(1)}$
30:     Update $\boldsymbol{\omega}$ using (17)
31:     Update $\psi^z$ using (16)
32: **end for**

---

## E.2  IMPLEMENTATION DETAILS

We used MAPPO as the backbone algorithm in updating c-MARL policies. Our implementation has been built on the implementation of HARL (Zhong et al., 2024) and EIR-MAPPO (Li et al., 2024). We used parameter sharing across the agents, thus we maintained a single belief network, a single c-MARL policy network, and one network per each adversarial type (in total $K$).

As baselines, we used EIR-MAPPO (Li et al., 2024), Generalized Maxmin (Gen-Maxmin) (Liu et al., 2024a), and RAP (Vinitsky et al., 2020). EIR-MAPPO can be regarded as a special case of BATPAL with only a single adversarial type, and we used their original implementation in our

Table 6: Hyperparameters used in all environments

| Hyperparameter | Value / Description |
|---|---|
| Discount factor ($\gamma$) | 0.99 |
| Actor network | MLP |
| Belief network | GRU |
| Belief hidden layer | single layer with 128 units |
| Policy learning rate ($\beta$) | 0.0005 |
| Adversary learning rate ($\alpha$) | 0.0005 |
| Critic learning rate | 0.0005 |
| Entropy coefficient | 0.01 |

Table 7: The values used as $V_{max}$ and $V_{min}^v$.

| Environment | $V_{max}$ | $V_{min}^v$ |
|---|---|---|
| **SMAC-2s3z** | 15.3 | 4 |
| **SMAC-MMM** | 16 | 10 |
| **SMAC-11m** | 16 | 10 |
| **SMAC-1c3s5z** | 16 | 7 |
| **LBF** | 0.55 | 0.3 |
| **MPE-Spread** | -50 | -130 |

comparisons. Gen-Maxmin models an adversary that at each time step in a two-agent setting, plays a worst-case attack (trained using adversarial learning) with probability $q$ and a cooperative policy with probability $1 - q$. We adapted their algorithm to the multi-agent setting and set $q = 0.5$. Moreover, based on the results reported by Liu et al. (2024a) we selected a learning rate of 0.0001 for non-victim agents and 0.0005 for the adversarial agents. RAP, on the other hand, considers a population of adversarial policies; however, unlike our method, these policies are not differentiated by behavioral diversity and are all trained under the max–min principle. RAP is originally designed for single-agent RL, and we adapted it to the c-MARL setting.

In our implementations, the value of log barrier coefficient $\lambda$ is 0.1 in SMAC environments and 0.2 in the other two environments. The rest of the hyperparameters are the same for all environments and are reported in Table 6. Moreover, the values used as $V_{max}$ and $V_{min}$ in all environments are shown in Table 7.

### E.3   COMPLEXITY CONSIDERATIONS

As mentioned earlier, we keep the complexity of BATPAL low by using parameter sharing and randomized updates of the adversarial network. In terms of the number of policy updates, this makes BATPAL comparable to standard adversarial learning. Our method does require pretraining a reference policy, but this stage uses far fewer samples than the full adversarial training, since the reference policy does not need to reach complete convergence. Another difference is that BATPAL requires two sets of experience samples (corresponding to the two POMDPs). However, sample collection is performed in parallel across multiple rollouts, and we handle this efficiently by splitting the rollouts into two groups and assigning one half to the POMDP associated with the reference policy experiences. To assess the overall complexity, Table 8 compares the execution time of training BATPAL with that of the baselines for the same number of environment time steps, corresponding to the results reported in Figure 2. The execution times are measured in terms of wall-clock time, and for a unified comparison, we normalize all times by the execution time required for the full convergence of vanilla MAPPO in the non-adversarial setting on the same hardware. By comparing the execution times in Table 2, we observe that although BATPAL has higher complexity than the baselines, the resulting increase in computation time is not significant, and is essentially the cost paid for achieving unified robustness.

From another perspective, with respect to $K$, one would expect that more samples and more policy model updates are required to train a policy that performs equally well against all adversarial types

Table 8: Comparison of the execution times. The execution times are normalized to the execution time of a vanilla MAPPO until convergence.

| Environment | BATPAL | EIR-MAPPO | Gen-Maxmin | RAP |
|---|---|---|---|---|
| **SMAC-2s3z** | 2.83 | 2.52 | 2.61 | 2.36 |
| **SMAC-MMM** | 3.53 | 3.21 | 3.26 | 3.16 |
| **LBF** | 2.44 | 2.10 | 2.27 | 1.98 |
| **MPE-Spread** | 4.21 | 3.54 | 3.90 | 3.28 |

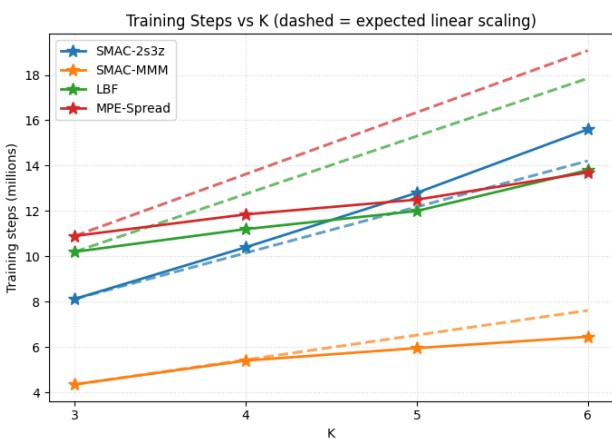

Figure 5: Training time-steps as a function of $K$. The dashed lines show the expected linear growth.

as $K$ increases. Given our randomized update of the networks, which corresponds to sampling different types during training, it is reasonable to anticipate that the number of samples needed to reach a given performance level would grow proportionally. For example, one may expect an increase by a factor of $5/4$ when $K$ increases from 3 to 4 (note that there is also a non-adversarial type). To assess this, we evaluated the number of environment time steps required to reach an average reward over all types equal to $90\%$ of the optimal non-adversarial policy. Figure 3 shows the results of this evaluation, and shows that except for SMAC-2s3z, which shows close to linear growth, the other environments remain well below the expected linear growth. As a conclusion, if a more complex environment requires finer severity levels, the increase in complexity with respect to $K$ can be expected to be sublinear.

