# OpenReview forum: "Bayesian Robust Cooperative Multi-Agent Reinforcement Learning Against Unknown Adversaries"
_ICLR.cc/2026/Conference — ICLR 2026 Poster_

### Official Review · Reviewer_JzM1 · 2025-10-15

**Soundness:** 3
**Presentation:** 2
**Contribution:** 3
**Rating:** 6
**Confidence:** 4

**Summary:**

This paper provides a method to optimize against Bayesian adverrsaries with several unknown types that corresponds to different reward reductions. The authors initially assume a continuous type space, then discretize it to some representitive type spaces, thus learning a more flexible policy against attackers. Experiments show higher robustness of their methods.

**Strengths:**

1. The paper solves the problem of continuous type in MARL with Bayesian attackers. While previous works solves the problem with 0-1 type, the paper expands the problem to continuous space and give solid theoretical derivations.

2. Technically, the authors propose BATPAL to gain robustness against varying attackers. While experiments are not highly extensive, it sufficiently verifies the claim of users.

**Weaknesses:**

1. only one adversary is considered. This is probably inherited from EIR-MAPPO, but this remains a limitation nonetheless. Since the attackers are less severe, maybe we can consider multiple attackers?

2. A very simple method of controlling the severity of attack is by a linear combination of a benign policy and an adversarial policy, as shown in [1]. Why we need to learn a constrained policy as shown in Section 4? I'm not asking for additional experiments, just curious about the design since current method is quite complex both theoretically and empirically, and I wonder if a simple interpolation of policies can solve most problems.

[1] Action Robust Reinforcement Learning and Applications in Continuous Control.

3. I do not fully understand Eqn. 7. Since it is proposed to solve (4), and in (4) the problem is formulated as a min-max game over a shared value function, then why we have two POMDPs? My understanding is we need to minimize reward while constraining it in a  certain range, but in this way it is a single constrained POMDP.

4. Do there exist an exact threat model of adversaries and a clear definition of adversaries? I see some definitions in Sec 4.1 but am curious of how  to formally determine a "ground truth" type given an attack policy. As also claimed by authors, two attack policies can vary in parameter and behaviors, while a single defense policy may not be effective for both.

5. In Proposition 3.2 the importance of generating diverse adversaries are proposed. However, in Eqn. 7, 8 and 9, I wonder which term corresponds to diversity. I assume it is related to \theta_1 and \theta_2 in Proposition 3.2? but numbers are different. What refers to 0 and 1 in subscripts with bracket?

6. Clarity. This is not a big issue, but I would recommend using bullet points or explicit subsections to improve the organization of the paper. Currently some important informations and settings are hard to find in a glance.

7.  Proposition 3.2 seems to be the difference between (1) and (2)? Guess this is a typo.

**Questions:**

See weakness.

---

> ### Author Response · Authors · 2025-11-26
> **Response to Reviewer JzM1 (part 1)**
>
> We thank the reviewer for the comments. Our responses are as follows:
>
> ## Weakness 1
> Our main reason for focusing on the single-victim setting is that we consider deployment-time attacks. To attack multiple agents simultaneously, an adversary would need to compromise multiple agents simultaneously, which may be significantly more challenging.
> However, our proposed method can be extended to the multi-victim scenario. For a coordinated attack on $N_v$ victim agents, $v_1,v_2,...,v_{N_v}$, we can treat the victim agents as a single “compound” agent whose action and observation spaces correspond to the joint actions and observations of all victim agents. This results in a system with $N-N_v$ benign agents, with some severity score, and one adversary. Our analysis applies in this setting, and BATPAL can be used for training. The main shortcoming of this approach is the combinatorial explosion of the sets of victim agents with increasing ${N_v}$, as  the prior distribution would have to be defined on all possible combinations of $N_v$ agents, which would also have to be explored during training.
>
> In principle, training a policy this way can provide robustness against attacks on any number of agents up to $N_v$, since, for example, an attack on $N_v - 1$ agents can be viewed as an attack on $N_v$ agents with lower severity. We added a section on this topic (Appendix D.5), and evaluated scenarios with two and three victims in the SMAC-MMM environment. Our results  show that BATPAL outperforms the baselines in this setting as well.
>
> ## Weakness 2
> The main issue with the approach mentioned by the reviewer is the lack of robustness of the trained policy to unseen attacks, similar to the training only against the worst-case attack, except that in this case the MARL policy becomes less robust to the worst case attack. In fact, one of our baselines (Gen-Maxmin) explicitly considers the linear combination of benign and adversarial policies suggested by the reviewer as the attack used during training. As shown in Figure 2, this approach provides some robustness against mild attacks, but it performs poorly  when facing the most severe ones.
> Our method, on the contrary, is consistently robust to all attack policies. It is a principled approach grounded in the notion of minimizing Bayesian regret. Intuitively, the constrained problem ensures that (1) the agent encounters diverse types of attacks, which are representative of the entire space of adversarial policies during training, and (2) the learned policy performs well against all of them in a Bayesian sense.
>
> **On the complexity of our method:** To keep the computational cost low in the implementation of BATPAL,  upon each update of the c-MARL policy, we only update one out of $K$ adversarial policies, chosen at random, rather than all $K$ adversarial policies. This design choice ensures that the computational cost, in terms of the number of policy model updates, is not significantly higher than that of standard adversarial learning (where a single adversarial policy is trained at the same time as the c-MARL policy). This design choice was included in our pseudo-code in the Appendix, but it was not sufficiently emphasized in the main text. In our revised manuscript we emphasize this choice in the evaluation section.
> We also added an evaluation of the execution time to the revised manuscript (Appendix E.3), including an evaluation of the execution time of our method and of the baselines, evaluated over the same number of environment time steps (corresponding to the results reported in Figure 2). The results are shown in Table 7, and show that the difference between the execution time of  BATPAL and that of the other baselines is not significant, i.e., BATPAL provides unified robustness at the cost of a small increase in computational complexity.

---

> > ### Author Response · Authors · 2025-11-26
> > **Response to Reviewer JzM1 (part 2)**
> >
> > ## Weakness 3 and Weakness 4
> > Our threat model and the definition of adversary types are presented in Sections 2.3 and 2.2, respectively. The type of an adversary can, in principle, be characterized by its private reward function. However, this reward function is unknown and cannot be used in practice. Instead, we define the ground-truth type of an adversary based on its performance when the benign agents follow a fixed reference policy $\boldsymbol{\pi}_0$. Note that the performance is expressed using the resulting expected initial value function (rather than the reward function). In other words, we rank all possible attacks based on their performance against $\boldsymbol{\pi}_0$ and then perform the discretization. What we seek in (5) (previously (4)) is to find a single policy that is robust against all types in expectation in a Bayesian sense.
> >
> > Equation (8) (previously (7)) then focuses on a specific discrete type, i.e., all adversarial policies whose reference value (against $ \boldsymbol{\pi}_0 $) lies within a constrained interval, and we seek the worst-case policy among them. However, “worst-case” here is measured against the actual robust policy of the non-victim agents $ \boldsymbol{\pi}^* $. In other words, the objective function uses the value with respect to $\boldsymbol{\pi}^*$ of the non-victim agents, while the constraint is defined using the value with respect to $\boldsymbol{\pi}_0$. Note that fixing the non-victim agents to any specific policy induces a particular POMDP from the adversary’s point of view. This makes the POMDPs in the objective and the constraint different. We revised the beginning of Section 4.1 to make this point clearer.
> >
> > ## Weakness 5
> > Proposition 3.2 (together with the newly added Proposition 3.3) characterizes the diversity of the policies obtained after finding the PBE. The equations mentioned by the reviewer in Section 4.1 pertain to the process of finding the PBE itself and are therefore not directly related to policy diversity. As explained above, the constraint and the objective function in our externally constrained RL problem correspond to different POMDPs, and indices 0 and 1 distinguish these two POMDPs.
> >
> > ## Weakness 6
> > We thank the reviewer for the suggestion. We applied it for organizing our evaluation section.
> >
> > ## Weakness 7
> > Yes, it is indeed a typo. We corrected that.

---

> > > ### Comment · Reviewer_JzM1 · 2025-11-27
> > > **Response**
> > >
> > > Thanks for the clarification. Now the paper seems clearer with corrected typos in some equations. I remain the score unchanged since we still do no know what is the optimal defense against unknown adversaries. However, I would say there may not be an explicit solution to this problem and the current version provides an acceptable approxmation. So I would  recommend accepting this paper.
> > >
> > > As a minor, I would also recommend checking the use of \citep and \citet, as some citations seems weird.

---

> > > > ### Author Response · Authors · 2025-12-02
> > > >
> > > > We thank the reviewer for the suggestion. We corrected the citations in the latest version of our submission.

---

### Official Review · Reviewer_GewT · 2025-10-30

**Soundness:** 2
**Presentation:** 2
**Contribution:** 3
**Rating:** 6
**Confidence:** 4

**Summary:**

This paper studies execution-time robustness in cooperative multi-agent reinforcement learning under unknown adversarial objectives. The authors formulate the problem as a Bayesian Dec-POMDP with a continuous adversarial type space and make it tractable by discretizing adversarial types based on performance degradation (severity) relative to a reference policy. The resulting framework seeks a Perfect Bayesian Equilibrium (PBE), where the cooperative policy conditions on beliefs over adversarial types. To generate representative adversaries, the paper proposes an Externally Constrained RL formulation solved via a log-barrier gradient method with convergence guarantees and introduces a practical EC-PPO variant for stability. A severity-dependent regret bound theoretically shows that finer partitions reduce regret and improve adaptivity. Empirical evaluations on SMAC, LBF, and MPE-Spread demonstrate consistent performance gains over baselines across both seen and unseen adversarial conditions.

**Strengths:**

1. Conceptual novelty and clarity of the discretized Bayesian game:
The paper presents a novel Bayesian framework that models adversarial uncertainty as a continuous type space and discretizes it based on the degree of performance degradation (severity). Here, severity is defined as the reduction in performance relative to a fixed cooperative reference policy, which allows the continuous adversarial type space to be transformed into a computationally tractable finite set of types, formulated as a Bayesian Dec-POMDP $\hat{M}_B$ (Eq. 2). The equilibrium concept of this discretized game is defined as a Perfect Bayesian Equilibrium (PBE), where the cooperative policy is conditioned on beliefs over adversarial types and adaptively responds to different levels of attack severity (Eq. 4). This formulation clearly captures cooperative–adversarial interactions under adversarial uncertainty, enabling tractable reasoning over continuous uncertainty without relying on manually defined attacker classes.

2. Technical contributions for adversary synthesis:
The paper introduces an Externally Constrained Reinforcement Learning formulation for training adversarial policies. Each adversary aims to minimize the cooperative team’s return while satisfying the severity constraint defined over the partitioned type space. To address this constrained optimization problem, the authors propose a log-barrier policy-gradient method and theoretically prove convergence to a KKT point under mild assumptions. Furthermore, to ensure practical stability, they propose the EC-PPO algorithm, which incorporates PPO-style clipping to alleviate instability near constraint boundaries. The algorithm performs simultaneous gradient updates between the cooperative agents and adversaries, allowing adversaries to act as approximate minimizing oracles while the cooperative policy progressively adapts to them.

3. Theoretical and Empirical Insights into Severity-Based Partitioning:
The paper proposes a severity-based partitioning of the adversarial type space to enhance the diversity of attack behaviors and provides both theoretical and empirical justifications for this design. This approach demonstrates that considering multiple severity partitions, rather than training a single worst-case (max–min) adversarial policy, leads to lower regret and higher adaptability, as mathematically established in the theoretical analysis. Such partitioning enables the model to learn representative adversarial policies for each severity level, thereby maintaining training stability and convergence while capturing a wider spectrum of adversarial strategies. Furthermore, experimental results presented in Section 5 (Figure 2) reveal that the most severe attacks do not always cause the greatest performance degradation. This finding suggests that the impact of adversarial behavior cannot be fully explained by attack strength alone, as attacks of different severity levels may exert distinct strategic effects. In particular, in the LBF, MPE-Spread, and SMAC-2s3z environments, moderate-intensity attacks were observed to disrupt cooperative balance more severely than the strongest attacks, which is consistent with the theoretical analysis. Overall, the proposed severity-based partitioning effectively handles continuous uncertainty in adversarial intensity and provides a principled means to achieve quantitative robustness across a diverse range of attack strengths.

**Weaknesses:**

1. Limited threat model and multi-agent realism:
The paper assumes a restricted threat model in which at most one victim agent is attacked per episode. While this setting enables analytical simplification (Sec. 3), it may not fully capture the diversity of adversarial scenarios that can arise in realistic multi-agent environments. For instance, the EGA mechanism in ROMANCE [1] generates a wide range of attack policies through evolutionary processes, and the Wolfpack attack in WALL [2] dynamically targets multiple agents across timesteps, reflecting more complex and coordinated adversarial behaviors inherent to multi-agent systems. To better account for such multi-agent adversarial settings, interactive and cooperative attack characteristics beyond those of single-agent scenarios may need to be further considered. The proposed method provides meaningful diversity by learning various single-victim attacks with different severities; however, this diversity appears to focus mainly on the single-agent perspective and might not fully capture the broader robustness challenges that could emerge from simultaneous or cooperative multi-agent adversarial interactions. Including a discussion or empirical extension on how the proposed Bayesian partitioning framework could be adapted or evaluated under these richer multi-agent adversarial scenarios would help clarify the scope and enhance the practical relevance of the work.

2. Limited diversity of experimental scenarios:
The experimental evaluation appears to be conducted on a relatively limited set of scenarios — one environment each for LBF and MPE, and two scenarios for SMAC. Compared to recent studies on robust MARL, which often consider a broader range of tasks or larger-scale multi-agent systems, the current evaluation setup may provide only a partial view of the proposed method’s generality. Expanding the experiments to include additional and more diverse cooperative environments (e.g., larger SMAC maps, alternative LBF configurations, or other benchmark multi-agent settings) would further enhance the empirical validation and offer a more comprehensive assessment of the method’s robustness and scalability.

3. Computational cost and scalability with respect to severity partitioning:
The proposed approach maintains a separate adversarial policy for each severity level K, suggesting that computational complexity may increase as K grows. In the appendix, results are reported only for K = 3, 4, 5, and the authors also note that training becomes more demanding for larger K. In more complex environments, finer partitioning could require learning multiple diverse adversarial policies, which may introduce scalability challenges in terms of training time and computational resources. Providing quantitative measurements of the computational cost analysis would enhance the empirical transparency and credibility of the study.


[1] Yuan, L., Zhang, Z., Xue, K., Yin, H., Chen, F., Guan, C., Li, L., Qian, C., and Yu, Y. Robust multi-agent coordination via evolutionary generation of auxiliary adversarial attackers. In Proceedings of the AAAI Conference on Artificial Intelligence, volume 37, pp. 11753–11762, 2023.
[2] Lee, Sunwoo, et al. "Wolfpack Adversarial Attack for Robust Multi-Agent Reinforcement Learning." International conference on machine learning. PMLR, 2025.

**Questions:**

Minor notation and typographical issues:

1.	In Equation (12), it is unclear what the subscript T in S_T specifically denotes. Given the surrounding context, it appears that this might be a typographical error and that the intended notation is S_t, representing the state at timestep t. Clarifying whether T refers to a specific terminal time or if it is indeed a typo would help prevent confusion.

2.	In Appendix Table 1, the entry labeled “SAMC-2s3z” seems to contain a typographical error; it should likely read “SMAC-2s3z.” Correcting this minor issue would improve the overall consistency of the presentation.

---

> ### Author Response · Authors · 2025-11-26
> **Response to Reviewer GewT**
>
> We thank the reviewer for the comments. Our responses are as follows:
>
> ## Weakness 1
> Our main reason for focusing on the single-victim setting is that we consider deployment-time attacks. To attack multiple agents simultaneously, an adversary would need to compromise multiple agents simultaneously, which may be significantly more challenging.
> However, our proposed method can be extended to the multi-victim scenario. For a coordinated attack on $N_v$ victim agents, $v_1,v_2,...,v_{N_v}$, we can treat the victim agents as a single “compound” agent whose action and observation spaces correspond to the joint actions and observations of all victim agents. This results in a system with $N-N_v$ benign agents, with some severity score, and one adversary. Our analysis applies in this setting, and BATPAL can be used for training. The main shortcoming of this approach is the combinatorial explosion of the sets of victim agents with increasing ${N_v}$, as  the prior distribution would have to be defined on all possible combinations of $N_v$ agents, which would also have to be explored during training.
>
> In principle, training a policy this way can provide robustness against attacks on any number of agents up to $N_v$, since, for example, an attack on $N_v - 1$ agents can be viewed as an attack on $N_v$ agents with lower severity. We added a section on this topic (Appendix D.5), and evaluated scenarios with two and three victims in the SMAC-MMM environment. Our results  show that BATPAL outperforms the baselines in this setting as well.
>
> ## Weakness 2
> In our original submission, we evaluated BATPAL on MMM with 10 agents, which to the best of our knowledge, is the largest SMAC environment considered in the context of robust c-MARL in the literature. To enhance the diversity of scenarios as the reviewer suggested, we evaluated two other SMAC scenarios, namely 1c3s5z and 11m, with 9 and 11 agents, receptively. The corresponding results are provided in Table 5, Appendix D.6, and show that BATPAL remains superior to the other baselines in providing consistent robustness, especially in SMAC-1c3s5z, where the performance gap between BATPAL and the other methods is substantial.
>
> ## Weakness 3
> To keep the computational cost low in the implementation of BATPAL,  upon each update of the c-MARL policy, we only update one out of $K$ adversarial policies, chosen at random, rather than all $K$ adversarial policies. This design choice ensures that the computational cost, in terms of the number of policy model updates, is not significantly higher than that of standard adversarial learning (where a single adversarial policy is trained at the same time as the c-MARL policy). This design choice was included in our pseudo-code in the Appendix, but it was not sufficiently emphasized in the main text. In our revised manuscript we emphasize this choice in the evaluation section.
>
> Conceptually, one would expect that more samples and more policy model updates would be needed to train a policy that performs well against various attacker types as well as in a non-adversarial setting, than a policy that performs well in a non-adversarial setting only. We performed additional evaluations to understand how the number of time steps, i.e., samples, required to reach an average performance equal to 90% of the non-adversarial setting, averaged over all types, depends on the number of attacker types $K$. Figure 5 in the Appendix shows the results, and indicates that the number of time steps required scales sub-linearly in $K$.
>
> In addition, to ensure scaling with respect to the number of agents, we employ parameter sharing, as noted in the evaluation section. Thus, the number of models is not affected by the number of agents. This is illustrated by Figure 1, which shows that doubling the number of agents (from 5 in SMAC-2s3z to 10 in SMAC-MMM) does not reduce the convergence rate.
>
> We added an evaluation of the execution time to the revised manuscript (Appendix E.3), including an evaluation of the execution time of our method and of the baselines, evaluated over the same number of environment time steps (corresponding to the results reported in Figure 2). The results are shown in Table 7, and show that the difference between the execution time of  BATPAL and that of the other baselines is not significant, i.e., BATPAL provides unified robustness at the cost of a small increase in computational complexity.
>
> ## Question 1
> Thank you for pointing out that. It should be $T_m$, i.e., the terminal time of episode $m$. We corrected this mistake.
>
> ## Question 2
> Thank you for pointing out the issue. We corrected it.

---

### Official Review · Reviewer_oTvg · 2025-10-31

**Soundness:** 3
**Presentation:** 2
**Contribution:** 3
**Rating:** 4
**Confidence:** 4

**Summary:**

This paper addresses the critical problem of robustness against unknown adversaries in cooperative Multi-Agent Reinforcement Learning (c-MARL). The authors propose BATPAL, a novel method that models the problem as a Bayesian Dec-POMDP with a continuum of adversarial types. To make the problem tractable, they introduce a partitioning scheme for the adversarial type space based on the severity of an attack, measured by its performance against a reference policy. This discretization allows them to frame the problem as finding a Perfect Bayesian Equilibrium in a finite-type game. Empirical results on several benchmarks demonstrate that BATPAL outperforms state-of-the-art baselines against a variety of attack strategies.

**Strengths:**

- Novel Problem Formulation: The Bayesian formulation to capture uncertainty over adversarial objectives is a relevant and timely direction for robust MARL, moving beyond the standard worst-case adversary assumption.

- Comprehensive Evaluation: The experimental section is thorough, using multiple diverse environments (SMAC, MPE, LBF) and a wide array of adversarial policies, including unseen dynamic adversaries, to evaluate the proposed method.

**Weaknesses:**

- Limited Theoretical Novelty and Justification for Partitioning: While the Bayesian game model is a good framing, the core partitioning mechanism, while intuitive, lacks a strong theoretical justification. Proposition 3.2 provides a bound on KL divergence, but this bound can be very loose in practice, and it is not conclusively demonstrated that this partitioning is the optimal or most efficient way to discretize the type space. The choice feels more heuristic than principled, and the theoretical benefits (mitigating local optima, ensuring diversity) are stated but not rigorously proven.

- High Complexity and Insufficient Ablation: The proposed method is significantly more complex than the baselines, requiring a pre-trained reference policy, K separate adversarial networks, a belief network, and a two-timescale update. This complexity raises concerns about scalability and computational cost. A major weakness is the lack of a proper ablation study. It is unclear how much each component (e.g., the Bayesian belief, the specific partitioning, the EC-PPO component) contributes to the final performance. For instance, would a simpler ensemble of adversaries trained with different fixed rewards perform similarly? The gains could be attributed more to exposure to a diverse set of adversaries during training rather than the specific Bayesian partitioning scheme.

- Technical Presentation and Proofs: The presentation of the externally-constrained RL algorithm and its convergence proof is highly technical and complex. While Proposition 4.2 claims convergence, the assumptions (e.g., perfect critics, bounded derivatives, a strictly feasible starting point) are very strong and often not met in deep RL practice. The subsequent switch to a more practical EC-PPO variant, while understandable, undermines the theoretical guarantees provided earlier. This creates a gap between the theory and the practical implementation.

**Questions:**

Please refer to the weakness

---

> ### Author Response · Authors · 2025-11-26
> **Response to Reviewer oTvg (part 1)**
>
> We thank the reviewer for the comments. Our responses are as follows:
>
> ## Weakness 1
>  Note that the type space is an unknown subset of $\mathbb{R}^{|S|\times|A^i|}$ (all possible adversarial rewards), which is equivalent to the set $\Theta^i = [0,1]$ (there is a bijection between these two sets).
> A main conceptual novelty introduced in our work is giving a meaning to the notion of adversarial type on $[0,1]$. The notion of optimality over the original type space is not well defined, while our proposed approach maps the original type space to a type-space that has a well-defined meaning through the reference-based severity definition. Our approach then discretizes the type space uniformly, which allows the training of robust policies.
>
> It is indeed an interesting question whether a better mapping exists, both in terms of assigning a meaning to the type-space $[0,1]$ (as an alternative to  severity) and in terms of the partitioning of that type space.
> For the latter, we do not claim that uniform discretization, which we have explored, is optimal. One could explore alternatives, e.g., ones that are more dense for high severity scores. One may also optimize the severity levels using gradient ascent by perturbing them individually, but doing so would lead to significant computational cost. Due to the involved computational cost, we choose to leave this investigation subject of future work. Nonetheless, we want to highlight that we are first in formulating an adversarial type space with a well defined meaning in this context.
>
> Moreover, it is important to note that the selection of representative policies for adversarial types, which is the key component of our proposed discretization, is not arbitrary. The consideration of the worst-case policy in each set follows directly from the notion of PBE. We demonstrated this in Appendix B.1 of the original submission, where we also showed that a robust policy constructed in this manner minimizes the Bayesian regret. We have revised our pointer to Appendix B.1 to make this argument clearer.
>
> We agree with the reviewer that the link between Proposition 3.2 and the diversity of adversarial polices generated by BATPAL was not clear in the original submission. Accordingly, in our revised submission, we define the diversity as
> $  Div(\lbrace \rho^{v,\theta_k} \rbrace_{k=1}^K)= \frac{1}{K(K-1)} \sum_{i\neq j}\mathbb{E}_s [ KL ( \rho^{v,\theta_i}(s) ||  \rho^{v,\theta_j}(s)) ] $,
> and in Proposition 3.3 we explicitly provide a lower bound on the diversity of adversarial policies generated by BATPAL as follows:
> $Div\geq \frac{(1-\gamma)^2(V{max}-Vmin)^2(K-2)}{12K}$
>
> We argue that this bound is not generally loose and, in fact, provides meaningful insight. The usefulness of the bound depends on the specific problem instance. If $V_{\text{max}}$ and $V_{\text{min}}^v$ are very close, then the bound becomes loose; however, this situation essentially indicates that adversarial policies have limited impact, and hence a large diversity of adversarial behaviors is unnecessary during training.
> On the contrary, in many practical scenarios, $V_{\text{max}}$ and $V_{\text{min}}^v$ can be as large as $\frac{1}{1-\gamma}$ and as small as $\frac{-1}{1-\gamma}$, respectively. In this setting, the right-hand side is close to $\frac{K-2}{3K}$, which is approximately 0.16 for $K=4$ and is bounded by $1/3$ as $K$ increases. As a baseline for comparison, observe that in TRPO [1], two policies with a KL divergence below 0.02 are considered "similar", whereas our bound is significantly larger. Finally, observe that this bound concerns the average distance between any two policies, rather than, e.g., the maximum distance, which would be much higher.
>
> [1] Schulman et al., "Trust Region Policy Optimization", 2015

---

> > ### Author Response · Authors · 2025-11-26
> > **Response to Reviewer oTvg (part 2)**
> >
> > ## Weakness 2
> > **On Complexity:** To keep the computational cost low in the implementation of BATPAL, upon each update of the c-MARL policy, we only update one out of $K$ adversarial policies, chosen at random, rather than all $K$ adversarial policies. This design choice ensures that the computational cost, in terms of the number of policy model updates, is not significantly higher than that of standard adversarial learning (where a single adversarial policy is trained at the same time as the c-MARL policy). This design choice was included in our pseudo-code in the Appendix, but it was not sufficiently emphasized in the main text. In our revised manuscript we emphasize this choice in the evaluation section.
> > One difference between BATPAL and a method like EIR-MAPPO, which also includes a belief network, is the need to pretrain the reference policy in our method. However, this pretraining stage requires far fewer samples than the full adversarial training (between 10% to 15%), since it is not necessary to train the reference policy to full convergence in practice.
> >
> > In our revised submission, we added a section on computational complexity  (Appendix E.3), and included an evaluation of the execution time of our method (including the pretraining) and of the baselines, evaluated over the same number of environment time steps (corresponding to the results reported in Figure 2). The results are shown in Table 7, and they show that the difference between the execution time of BATPAL and the baselines is not significant, i.e., BATPAL achieves significantly improved robustness at the price of a small increase in computational cost compared to the baselines.
> >
> >  **On Ablation Study:** We added an ablation study to the revised manuscript (Section 5.2) where we consider BATPAL without a belief and with a perfect belief, an alternative policy gradient method in place of EC-PPO, and a training scheme against an ensemble of  adversaries. The results in Table 1 show that removing the belief network or using an alternative algorithm in place of EC PPO can lead to significantly lower performance against unseen attacks. For the last case, i.e., training against a fixed set of diverse adversaries, we used $K=4$ adversarial policies obtained based on the DYN rewards defined by (Kazari et al., 2023), each corresponding to a different severity level. Table 1b shows that this approach leads to significantly lower performance compared to BATPAL, for example in SMAC-2s3z. The result shows that while diversity is necessary, it is not sufficient to achieve generalizable robustness against unknown adversaries. Another evidence for this claim is the RAP baseline. RAP also trains c-MARL against a population of adversaries, where diversity is achieved through different instantiations of the learning process rather than through different objectives. However, it can be observed that RAP does not either provide the consistent robustness that BATPAL achieves.

---

> ### Author Response · Authors · 2025-11-26
> **Response to Reviewer oTvg (part 3)**
>
> ## Weakness 3
> The notion of a “perfect critic” we used in the original submission may have been somewhat misleading, we apologize for that. Our proof does not require a perfect critic; it is sufficient to assume unbiased estimates of the value function. We revised this assumption in the revised manuscript, thus the assumptions in Proposition 4.2 are the standard assumptions in the RL literature:
> 1)  Unbiased estimates of values and gradients is standard in convergence analysis of deep RL [2], [3], [4]. It relies on the idea that the critic updates on a faster timescale than the policy, and can therefore be treated as approximately stationary. Without this assumption, the non-stationarity introduced by continuously updating the critic makes the theoretical analysis significantly more complex, which is beyond the scope of our work.
>
> 2) Bounded derivatives or log-gradients is a very common assumption in theoretical analysis of RL [4], [5], [6], and is needed for controlling the stochastic approximation process.
>
> 3) Assuming a feasible starting point is widely used in the analysis of safe RL [7]. Intuitively, if the feasible set is empty then no algorithm can be expected to produce a valid solution to a constrained RL problem, and the assumption about the starting point being feasible is needed due to barrier method used in the proof. In practice, even if we begin from an infeasible point, the reverse-gradient mechanism (the second line in (14) (previously (13))), can be used to move the iterate into the feasible region. Once this occurs, the assumption is satisfied from that point onward.
>
> [2]  Paternain et al., "Stochastic Policy Gradient Ascent in Reproducing Kernel Hilbert Spaces", IEEE Transactions on Automatic Control, 2020
>
> [3] Graves et al., "Off-Policy Actor-Critic with Emphatic Weightings", JMLR, 2023
>
> [4] Sutton et al., "Policy Gradient Methods for Reinforcement Learning with Function Approximation", NeurIPS, 1999
>
> [5] Konda et al., "Actor-critic algorithms", NeurIPS, 1999
>
> [6] Huang et al., "Occupancy-based policy gradient: Estimation, convergence, and optimality", NeurIPS, 2024
>
> [7] Yu et al., "Convergent policy optimization for safe reinforcement learning", NeurIPS, 2019

---

### Official Review · Reviewer_PTA9 · 2025-11-01

**Soundness:** 3
**Presentation:** 3
**Contribution:** 3
**Rating:** 6
**Confidence:** 3

**Summary:**

This paper proposes BATPAL, a Bayesian framework for making c-MARL robust against unknown adversarial behaviors. Instead of assuming a single worst-case adversary, the authors model uncertainty through a Bayesian Dec-POMDP with a range of adversarial types. The authors introduce an externally constrained RL formulation with the EC-PPO algorithm to learn representative adversarial policies, and use simultaneous gradient updates to train robust cooperative agents toward a Perfect Bayesian Equilibrium.

**Strengths:**

- The paper is extending standard MARL robustness beyond worst-case adversaries, which could potentially provide a new problem setup.
- The paper provides a theoretical analysis, such as regret bounds and a convergence proof for EC-PPO.

**Weaknesses:**

- Even though the method is theoretically sound,  the computational cost of training multiple adversarial policies (one per severity level and victim) could potentially be prohibitively large for with increasing agent team size.
- There seems to missing ablation such as the performance of the proposed method with and without the belief network, and how does the method performs with different under different accuracy of the blief network?

**Questions:**

- The paper uses 4 levels of severity. How does it affect the performance of the proposed method if we use a different number of severity levels?
- How does the EC PPO compare to other constrained RL algorithms? Are they any evaluation specifically on the model?
- What is the real-world applicability of the proposed method and what are the real-world use cases for this type of method where you can assume

---

> ### Author Response · Authors · 2025-11-26
> **Response to Reviewer PTA9 (part 1)**
>
> We thank the reviewer for the comments. Our responses are as follows:
>
> ## Weakness 1
> To keep the computational cost low in the implementation of BATPAL,  upon each update of the c-MARL policy, we only update one out of $K$ adversarial policies, chosen at random, rather than all $K$ adversarial policies. This design choice ensures that the computational cost, in terms of the number of policy model updates, is not significantly higher than that of standard adversarial learning (where a single adversarial policy is trained at the same time as the c-MARL policy). This design choice was included in our pseudo-code in the Appendix, but it was not sufficiently emphasized in the main text. In our revised manuscript we emphasize this choice in the evaluation section.
>
> Conceptually, one would expect that more samples and more policy model updates would be needed to train a policy that performs well against various attacker types as well as in a non-adversarial setting, than a policy that performs well in a non-adversarial setting only. We performed additional evaluations to understand how the number of time steps, i.e., samples, required to reach an average performance equal to 90% of the non-adversarial setting, averaged over all types, depends on the number of attacker types $K$. Figure 5 in the Appendix shows the results, and indicates that the number of time steps required scales sub-linearly in $K$.
>
> In addition, to ensure scaling with respect to the number of agents, we employ parameter sharing, as noted in the evaluation section. Thus, the number of models is not affected by the number of agents. This is illustrated by Figure 1, which shows that doubling the number of agents (from 5 in SMAC-2s3z to 10 in SMAC-MMM) does not reduce the convergence rate.
>
> We added an evaluation of the execution time to the revised manuscript (Appendix E.3), including an evaluation of the execution time of our method and of the baselines, evaluated over the same number of environment time steps (corresponding to the results reported in Figure 2). The results are shown in Table 7, and show that the difference between the execution time of  BATPAL and that of the other baselines is not significant, i.e., BATPAL provides unified robustness at the cost of a small increase in computational complexity.
>
> ## Weakness 2
> We added an ablation study to the revised manuscript (Section 5.2), including results without a belief network and with a perfect belief. The results show the importance of the belief network. Without the belief, the robustness of the c-MARL policy trained using BATPAL is significantly worse than when using the belief. Interestingly, when the perfect belief is fed to the policies, robustness still suffers (e.g. when facing attacks unseen during training, even though the belief captures the severity level of the attack). These results clearly show the importance of the belief. The ablation study we added also includes results for an alternative to EC-PPO, as well as results obtained using training against an ensemble of adversarial policies.

---

> > ### Author Response · Authors · 2025-11-26
> > **Response to Reviewer PTA9 (part 2)**
> >
> > ## Question 1
> > Appendix D.2 of the original submission provides results for different values of $K$. The results show that increasing $K$ improves the performance of BATPAL, but with a decreasing marginal gain. Therefore, a small $K$ is sufficient to achieve good performance in most cases.
> >
> > ## Question 2
> > As discussed in Section 4.1, there is a fundamental difference between standard constrained RL and our externally constrained RL problem. In our setting, the constraint and the objective are evaluated on different trajectories, whereas standard constrained RL assumes they are evaluated on the same trajectory. The common approach in constrained RL algorithms is to construct a new reward function as  a linear combination of the objective  and the constraint, and train a policy that maximizes this new reward. Such an approach is, however, not feasible in our problem due to the different trajectories, and therefore these algorithms are not directly comparable with EC PPO. To illustrate this, we included a modified version  of BATPAL in Section 5.2 of the revised manuscript, in which EC PPO is replaced by a vanilla policy gradient algorithm. From the results (Table 1), we observed that this variation is not generally robust against attacks with different adversarial types. This can be explained by the fact that, without the clipping mechanism, the policies can easily move far outside the feasible region. Consequently, the adversarial policies encountered during training are not representatives of their designated types, which undermines overall robustness.
> >
> > ## Question 3
> > For example, in multi-agent robotic tasks such as warehouse logistics or search-and-rescue operations, the system can be vulnerable to robot hijacking or sensor data perturbations. As a result, an affected agent may fail to cooperate with others in completing the task. Our method can be used to strengthen the system’s robustness, prevent cascading failures among robots or drones, and maintain cooperative task performance even when some agents are under attack.
> > Another example involves the erratic behavior of autonomous vehicles in multi-vehicle driving scenarios (e.g., caused by failures or by human intervention). Applying our method can yield policies that preserve coordination despite such disturbances.

---

### Author Response · Authors · 2025-11-26
**Overview of Updates in the Revised Manuscript**

We thank the reviewers for the insightful feedback. We have revised the manuscript to address all comments, by improving clarity, by providing theoretical justification, and by extending the experimental validation. Below we provide a summary of the main changes compared to the original submission, together with the corresponding comments from the reviewers, shown in **bold font**.

1) Theoretical justification (**oTvg(W1)**): To enhance the theoretical justification for using BATPAL, we added a new proposition (Proposition 3.3), which provides a lower bound on the diversity of adversarial policies obtained by BATPAL. This proposition, together with Appendix B.1 and propositions 3.1 and 3.4, supports that BATPAL has a good theoretical foundation.

2) Ablation study (**PTA9(W2, Q2), oTvg(W2)**): We added an ablation study (Section 5.2) demonstrating the importance of the different components in our method. It includes evaluations of BATPAL without a belief network, with a perfect belief, with an alternative policy gradient method in place of EC-PPO, and with a training scheme that uses an ensemble of adversaries. The results demonstrate that all components of BATPAL contribute to its robustness.

3) Evaluation (**GewT(W2)**): To increase the diversity of evaluated environments, we added results for two additional SMAC environments, namely 1c3s5z and 11m, in Appendix D.6. These are among the largest SMAC environments and illustrate that the superiority of our method over the baselines scales to more complex settings.

4) Computational burden (**PTA9(W1), oTvg(W2), GewT(W3)**): We added a section on complexity considerations (Appendix E.3), showing that the increase in execution time relative to other baselines is not significant (Table 7), while BATPAL achieves substantially better adversarial robustness. We also show empirically that the number of training time steps required for BATPAL increases sub-linearly with respect to $K$ (Figure 5). A key factor enabling low complexity is the randomized update of adversarial policies. In the original submission, this detail could only be inferred from the pseudo code in the appendix. In the revised manuscript, we clarify this point explicitly in the evaluation section.

5) Multiple victim agents (**GewT(W3), JzM1(W1)**): We added a section on adapting BATPAL to the case of multiple victim agents (Appendix D.5) and provided an empirical evaluation demonstrating its superiority over the baselines in this setting.

6) Clarity: We revised the text to improve clarity, in response to the reviewer comments.

---

### Meta-Review · Area_Chair_2YPk · 2025-12-20

**Summary:**

The reviewers pointed out several weaknesses and shared common concerns, including (a) potentially high computational cost of training, especially when agent team size grows; (b) missing ablation experiments and lack of diverse experimental scenarios; (c) restricted threat model and limited multi-agent realism; (d) limited theoretical novelty and justification for the proposed method; (e) large gap between the theoretical setup and proposed practical implementation.

**Reviewer Concerns:**

Several of the common concerns related to high computational cost, limited theoretical justification for the proposed method, and restricted threat model seem to be outstanding. The justifications provided in the rebuttal may only partially address the raised concerns.

**Reviewer Scores:**

Reviewer PTA9 would have likely kept their score at 6.

Reviewer oTvg would have likely kept their score at 4.

Reviewer GewT would have likely kept their score at 6.

Reviewer JzM1 would have likely kept their score at 6.

---

### Decision · Program_Chairs · 2026-01-26

Accept (Poster)